# Advances in Electrochemical Detection Electrodes for As(III)

**DOI:** 10.3390/nano12050781

**Published:** 2022-02-25

**Authors:** Haibing Hu, Baozhu Xie, Yangtian Lu, Jianxiong Zhu

**Affiliations:** 1Academy of Opto-Electric Technology, Special Display and Imaging Technology Innovation Center of Anhui Province, National Engineering Laboratory of Special Display Technology, State Key Laboratory of Advanced Display Technology, Collaborative Innovation Center of Advanced Display Technology, Anhui Key Laboratory of Advanced Imaging and Display Technology, Opto-Electric Display Industry Innovation Center, Anhui Province Key Laboratory of Measuring Theory and Precision Instrument, School of Instrument Science and Optoelectronics Engineering, Hefei University of Technology, Hefei 230009, China; xie_bz@163.com (B.X.); hfut_lyt@163.com (Y.L.); 2School of Mechanical Engineering, Southeast University, Nanjing 211189, China

**Keywords:** As(III) detection, electrochemical sensing, nanosensing, biosensing

## Abstract

Arsenic is extremely abundant in the Earth’s crust and is one of the most common environmental pollutants in nature. In the natural water environment and surface soil, arsenic exists mainly in the form of trivalent arsenite (As(III)) and pentavalent arsenate (As(V)) ions, and its toxicity can be a serious threat to human health. In order to manage the increasingly serious arsenic pollution in the living environment and maintain a healthy and beautiful ecosystem for human beings, it is urgent to conduct research on an efficient sensing method suitable for the detection of As(III) ions. Electrochemical sensing has the advantages of simple instrumentation, high sensitivity, good selectivity, portability, and the ability to be analyzed on site. This paper reviews various electrode systems developed in recent years based on nanomaterials such as noble metals, bimetals, other metals and their compounds, carbon nano, and biomolecules, with a focus on electrodes modified with noble metal and metal compound nanomaterials, and evaluates their performance for the detection of arsenic. They have great potential for achieving the rapid detection of arsenic due to their excellent sensitivity and strong interference immunity. In addition, this paper discusses the relatively rare application of silicon and its compounds as well as novel polymers in achieving arsenic detection, which provides new ideas for investigating novel nanomaterial sensing. We hope that this review will further advance the research progress of high-performance arsenic sensors based on novel nanomaterials.

## 1. Introduction

In China and the rest of the world, environmental pollution has always been an urgent problem that threatens the living environment and health of human beings. A variety of forms of environmental pollution exist at the same time, including soil, air, radioactive elements, water pollution, and so on. Among the most direct and obvious causes of harm to human beings, the heavy metal pollution of water environments, the toxicity of heavy metal-like arsenic (As) ions, and the degree of difficulty in detecting and removing them constitutes one of the most important forms of environmental pollution. Arsenic is an element that is widely distributed in soil, minerals, the aquatic environment, and the atmosphere. In terms of arsenic abundance, it ranks 20th in the Earth’s crust, 14th in seawater, and 12th in human systems when comparing all elements [1]. Widespread in nature, arsenic and its compounds are mobile in the environment and, when dissolved in the water species of arsenic, contamination will even enter the biosphere through the food chain. As early as the 1970s, in a study by the U.S. National Pesticide Monitoring Program targeting the detection of mercury, arsenic, lead, cadmium, and selenium residues in fish, the results show that more than 95% of the combined samples have detectable residues of all metals. This number is increasing year by year [2]. In natural water bodies, arsenic is usually present as arsenite (As(III)), arsenate (As(V)), monomethylarsenic acid (MMA), and dimethylarsenic acid (DMA). Inorganic forms of arsenite (As(III)) and arsenate (As(V)) have stronger circulation and are more likely to be enriched in the food chain, which is the main form of arsenic present in nature and has a great impact on human survival.

Heavy metals are a very important class of carcinogens, especially nickel, chromium, and arsenic, which are recognized as human carcinogens [3]. Radon and arsenic exposure were recognized as major risk factors in a cohort of Chinese tin miners with lung cancer [4]. Additionally, in China, Y J Lu et al. studied mineral dust deposition in the lungs of tin miners in Yunnan Province, and data obtained in lung tissue provided evidence for arsenic as a cause of lung cancer [5]. Exposure to inorganic arsenic in drinking water has the potential to cause changes in left ventricular geometry and cardiomegaly in adult males [6]. There are many ways for humans and animals to be exposed to arsenic (mainly through drinking water and the accumulation of arsenic in the food chain). Long-term exposure to arsenic ions may lead to a series of physiological diseases, including neurological, physiological, reproductive, kidney, liver, and even genetic conditions [7]. There is serious arsenic contamination of water and soil in many countries around the world, including China, and there was a large-scale arsenic ion poisoning incident in Bangladesh [8,9].

It is on account of these characteristics of arsenic that the detection of trace amounts of arsenic is very important. As early as the nineteenth century, John Bostock realized the importance of detecting traces of arsenic and observed different methods of doing so [10]. In 1901, S Delepine applied arsenic detection technology to life, detecting arsenic in beer and brewing materials [11]. In the following years, various techniques were applied to arsenic detection, and these techniques are divided into two main categories according to the instruments used: one is the use of traditional instrumentation techniques, including atomic absorption spectrometry, atomic emission spectrometry, inductively coupled plasma mass spectrometry, etc. The second is the use of new sensor methods, mainly biosensor methods, electrochemical detection, and the ultraviolet–visible spectroscopy (UV–Vis), colorimetric method, etc. In this paper, electrochemical detection methods will be introduced. Electrochemical detection has a series of advantages such as simple instrumentation, simple operation, high sensitivity, high selectivity, being easy to carry, and being easy to use for rapid detection. Electrochemical detection methods are an effective alternative to traditional ion detection methods.

## 2. Principles of Electrochemical Detection of As(III)

Probably one of the oldest measurement techniques, the electrochemical method, is an analytical method based on electrochemical principles and founded on the electrochemical properties of a substance in solution, which implies the transfer of charge between the electrode and the liquid or solid phase [12]. During the detection, the solution to be tested is usually used as part of a chemical cell, which reacts to the concentration of the substance being detected by means of the electrical parameters of the chemical cell. Compared with other arsenic detection methods, electrochemical detection is considered to be user-friendly and achieves a more desirable detection result, while the detection process is simple and cost effective. After a long period of development, electrochemical detection techniques to detect arsenic have become more mature, and there are many commonly methods based on various electrical signals used in the detection of heavy metals. Electrochemical techniques are divided into amperometric, voltammetric, potentiometric, impedance measurements, coulometric, and electrochemiluminescent techniques [13]. The general apparatus for electrochemical detection is shown in Figure 1 and usually consists of an electrolytic cell in which the heavy metal ions or other classes of heavy metal ions to be detected act as the electrolyte. A chemical reaction occurs in the cell, partly at the working electrode, resulting in a change in the electrical parameters that establish a link with the concentration of the target element for the purpose of quantitative analysis.

Voltammetry is considered the most versatile of the arsenic ion assays and has been further advanced with the development of potential and current modulation techniques, such as differential pulse voltammetry (DPV), square-wave voltammetry (SWV), stripping chronopotentiometry (SCP), etc. Voltammetry is based on the voltage–current time relationship generated in a three-electrode cell; the position of the peak is reflective of the specific chemical, and the density of the peak is related to the concentration of the substance being detected. The combination of electrode modification and pre-concentration techniques further increases the selectivity and sensitivity of voltammetry.

## 3. Electrochemical Detection Electrodes for As(III)

During decades of research, electrodes have often been modified in order to improve the sensitivity and selectivity of the electrochemical detection of ions. In the beginning, the excellent electrical properties of precious metals (e.g., gold and silver) were taken into account and a large number of electrodes made of precious metals were used for electrochemical detection. Later, there were considerations of cost constraints, reducing the amount of precious metals, and modifying the working electrodes with precious metals and their nanoparticles to achieve the same detection results. To further reduce costs, a range of materials including carbon nanomaterials, non-precious metal oxides, and bimetallic nanoparticles have been used for detection, again well below the WHO (World Health Organization) detection limit of 10 ppb. Additionally, in today’s interdisciplinary world, there are many biotechnologies used for electrochemical detection, with examples of bioreceptors such as DNA or proteins being used for modified electrodes.

### 3.1. Precious Metal Electrodes and Precious-Metal-Modified Electrodes

The excellent electrical properties of precious metals with good conductivity make them the preferred choice for electrode materials and their modifications in electrochemical detection. They show distinct advantages over the conventional macroelectrodes, such as increased mass transport, decreased influence of the solution resistance, low detection limit, and better signal-to-noise ratio [14]. In addition, precious metal particles can easily be deposited onto the electrode surface by electrochemical means, thus modifying the electrode and improving its overall surface properties.

#### 3.1.1. Gold Electrodes and Gold-Modified Electrodes

Gold electrodes in various forms and gold-modified materials have long been a popular topic in electrochemical detection. Agnese Giacomino et al. achieved a low detection limit of 0.060 ppb in the range of 1–15 ppb for the determination of As(III) by anodic dissolution voltammetry using a lateral gold electrode [15]. The electrochemical behavior of gold electrodes is closely related to their crystal orientation, and single-crystal Au(111) electrodes with clean, well-defined, and ordered surfaces can provide a more defined electrochemical behavior for As detection and are suitable for studying deposition mechanisms. Mohammad Rezaur Rahman et al. reported a simple method for the fabrication of Au(111)-like poly-gold electrodes, achieving a low detection limit of 0.28 ppb by square-wave anodic solvation voltammetry (SWASV) [16]. In addition, there are different sizes and forms of gold electrodes, including solid-disc gold electrodes [17], thin-film gold electrodes on glass carbon electrodes [18], graphite electrodes [19], platinum electrodes [20], microfilament gold electrodes [21], etc. Lijuan Bu et al. proposed the first method to electrically generate H_2_ to reduce As(III) and improve As(0) preconcentration on a gold disc electrode for the determination of As(III) by anodic stripping voltammetry, which improved the sensitivity of As(III) detection and achieved a detection limit of 1.0 nM or 0.075 ppb [22]. Gold wire microelectrodes are also a common class of gold electrodes and ChiZhou was able to detect As(III) down to 2.6 ppb in 0.5 M H_2_SO_4_ by square-wave anodic dissolution voltammetry using electrochemically etched gold wire microelectrodes [23]. S. Laschi et al. tested a disposable gold screen-printed working electrode for arsenic detection in aqueous solutions using square-wave anodic dissolution voltammetry (SWASV) and achieved a detection limit of 2.5 ppb after 60 s of deposition [24]. A self-made nanoporous gold microelectrode by Darío Xavier Orellana Jaramillo et al. had a high sensitivity of 29.75 μA (μg L^−1^)^−1^ cm^−2^ and a low detection limit of 0.62 ppb [25]. More comparisons of different gold-based electrodes are given in Table 1.

Additionally, due to the detection cost of using gold as an electrode, there has been increasing amounts of research on gold nanoparticle-modified conventional macroscopic electrodes, such as modified graphite electrodes [19], glassy carbon electrodes [40], screen-printed electrodes [41], gold electrodes [17], solid carbon paste electrodes [42] or boron-doped diamond electrodes [43], etc. Forsberg et al. tested three different materials (platinum, mercury, and gold) for modified electrodes [44], using anodic solvation voltammetry (ASV) and differential pulse anodic solvation voltammetry (DPASV), and found that the gold-modified electrodes were the most sensitive to the electrical signal generated by arsenic oxidation (which is one of the reasons why gold nanomaterials became the first material for modified electrodes). In addition to various forms of nanogold, other materials of composite-modified electrodes for the electrochemical detection of arsenic are a hot topic of research.

Supunnee Duangthong et al. developed a flow injection differential pulsed anodic dissolution voltammetry (FI-DPASV) method for the detection of arsenic, using a gold film-modified glassy carbon electrode as the working electrode and optimizing the parameters to achieve a low detection limit of 0.81 ppb in the linear interval 1.0–30 ppb for As(III) [45]. Syeda Sara Hassan et al. synthesized gold nanoflower structures by heating a mixture of ibuprofen and gold chloride (HAuCl_4_) at a constant temperature for about 30 min. The SEM images are shown in Figure 2. Ibu-AuPNFs modified screen-printed electrodes, followed by Nafion as a binder and stabilizer, were used in the 0.1–1800 ppb range, observed for As(III) with a linear calibration plot with a lower limit of detection of 0.018 ppb [46]. Tran Ngoc Huan proposed a three-dimensional (3D) gold (Au) nanodendritic network porous structure constructed by a simple electrochemical synthesis method, the SEM images of which in different views are shown in Figure 3, allowing for the more sensitive detection of As(III) due to its larger surface area [47]. Dao Anh Quang successfully synthesized and stabilized gold nanorods (GNR) using cetyltrimethylammonium bromide (GNR-CTAB) and poly(sodium-4-styrenesulfonate) (GNR-PSS); the TEM image and the diameter and length distribution of the synthesized nanorods are shown in Figure 4. The GNR-modified glassy carbon electrode showed an excellent response with a limit of detection (LOD) of 0.72 ppb and a linear concentration of As(III) between 0.90 and 38.99 ppb [48]. Dingnan Lu et al. reported a square-wave anodic dissolution voltammetry (SWASV) method using a new gold nanostar-modified screen-printed electrode (AuNS/SPCE) as the working electrode; the TEM image of the gold nanostar is shown in Figure 5. Electrochemical impedance spectroscopy tests showed that the charge transfer resistance of AuNS/SPCE (0.8 kΩ) was significantly lower compared with bare SPCE (2.4 kΩ), achieving a sensitivity of 0.2213 μA/ppb in the linear detection interval of 0–100 ppb for As(III) detection [49].

Among the many gold nanomaterials, gold nanoparticles and composites of gold nanoparticles on various materials are the absolute favorites in the field of electrochemical detection.

Anamarija Stanković et al. modified glassy carbon electrodes with gold nanoparticles and crystalline violet [50]. The electron impedance spectroscopy results show that the electron transfer resistance of the nanogold crystalline violet film was lower than that of the bare glassy carbon electrode, thus enhancing the electron transfer kinetics. The sensitivity of the modified electrode for the detection of As(III) was 5.6 A/μM cm^2^, the detection limit was 0.8 μM, and it had a good linear response in the range of 2.0–22.0 μM. Connor Sullivan et al. used a screen-printed electrode modified with nanogold to detect As(III) in commercial apple juice by square-wave dissolution voltammetry. The sensitivity of the method was 0.1007 μA ppb^−1^ and the detection limit was 16.73 μg L^−1^. The results of voltammetry were compared with those of graphite furnace atomic absorption spectrometry, with no systematic deviation and an R^2^ of 0.939 [51]. In recent years, electro-membrane extraction (EME) and anodic stripping voltammetry (ASV) combined with gold nanoparticle-modified glassy carbon electrodes (AuNPs/GCE) have also been used to detect As (III) in water, with a detection limit of 0.18 μg L^−1^ for this method [52]. The application of electromagnetic radiation before electrochemical determination improves the selectivity and cleaning ability of the sample, which can increase the lifetime of the working electrode and reduce surface passivation.

A simple and easy method for synthesizing bentonite (bt) clay-supported gold nanoparticles (AuNPs) composites has been reported [53], and the test plots are shown in Figure 6. The AuNPs were successfully synthesized and doped into bt clay, as shown by spectroscopic, microscopic and electrochemical methods, and the synthesized Au-bt material was used to modify the glassy carbon electrode (GC). The GC/Au-bt electrode was used to detect As(III) in neutral solution by cyclic voltammetry. The GC/Au-bt electrode showed a wide linear range with good reproducibility and stability in As(III) solutions in the range of 1–1700 μM. The detection limit was 0.1 μM with high sensitivity. In addition, it had good selectivity for the determination of As(III) in the presence of Cu(II) and other interfering ions, providing an effective new route for the measurement of As(III) under neutral conditions. Qian Tang et al. prepared Au-PANI-Fe-CNFs composites by forming polyaniline (PANI) nanosheet arrays on Fe-CNFs substrates followed by the self-deposition of Au nanoparticles; the process is shown in Figure 7, using the composite-modified glassy carbon electrode as a sensing platform for As(III) measurements [54]. Polyaniline showed a uniform array structure on the surface of carbon fibers (CNFs), and the presence of Fe in CNFs promoted the formation of polyaniline nanosheets and the adsorption of As(III) in the subsequent sensing process. The detection of As(III) in water has good electrochemical performance, and the SWASV response plot is shown in Figure 8. The electrode has a wide linear range (5–400 ppb) and high sensitivity with a detection limit of 0.5 ppb, which provides a new route for the electrochemical analysis of arsenic in water.

Besides composites of gold nanomaterials with carbon materials, metal oxide nanomaterials and certain polymers have also received much attention. More pairs of gold nanomaterial-modified electrodes are shown in Table 2.

#### 3.1.2. Platinum Electrodes and Platinum-Modified Electrodes

In addition to gold, there are other precious metals used in the electrochemical detection of arsenic. Platinum wires are not only used as counter electrodes in many electrochemical detection systems, but there are also many working electrodes made of platinum or modified with platinum nanoparticles. Tung Son Vinh Nguyen et al. determined the As(III) ion concentration in water samples by the anodic solvation voltammetry (ASV) technique using a Nafion film-modified platinum electrode and obtained a wide linear range from 0 ppb to 40 ppb with a detection limit below 10 ppb [102]. He Xu, on the other hand, used a Pt nanotube array electrode, the SEM in cross-sectional view of which is shown in Figure 9, achieving a low detection limit of 0.1 ppb. Electrochemical experiments demonstrated that platinum-nanotube array electrodes (PtNTAEs) exhibited better performance for As(III) analysis than Pt nanoparticle-coated GCE (Pt nano/GCE) or Pt foil electrodes [103]. In addition, the modification of other electrodes with Pt nanoparticles and their composites is also a promising approach [104,105,106,107,108]; the detection limit is mostly below 10 ppb. Of interest is a Pt single-atom-anchored catalyst on MoS_2_ (Pt_1_ /MoS_2_) developed by Pei-Hua L et al. to catalyze the determination of As(III). Pt_1_/MoS_2_ of 4% exhibited excellent stability and interference resistance, with a sensitivity of up to 3.31 μA/ppb for the detection of As(III) under near-neutral conditions, due to the Pt single atoms activated close to S atoms, greatly enhancing the catalytic activity of S atoms in the MoS_2_ plane [109]. Dong-Dong Han et al. demonstrated a size-dependent effect of 2–5 nm Pt nanoparticles on the electrochemical behavior of arsenic (As(III)), with a decrease in sensitivity as the size of Pt nanoparticles increased from 2.3 nm to 5.5 nm [110].

#### 3.1.3. Silver Electrodes and Silver-Modified Electrodes

Several papers have reported on the silver-based electrochemical sensing of arsenic. Among them, María del C. Aguirre used a silver wire electrode (SWE) for the electrochemical detection of arsenite with a minimum detection limit of 0.09 ppb, indicating that SWE can be used for the detection of trace arsenic in alkaline and neutral media [111]. Baudelaire Matangouo Sonkoue generated silver nanocolloids by the chemical reduction of silver salts using citrate in aqueous solution and used a gold electrode modified with silver nanoparticles as the working electrode in linear sweep voltammetry for the detection of arsenic ions. Under optimal conditions, calibration curves were plotted over a concentration range of 0.05–0.2 μM and the limit of detection was estimated to be 13.8 nM [112]. In addition, composites of silver nanomaterials with chitosan and graphene are also of interest for the electrochemical detection of arsenic. Silver nanoparticles (AgNPs) with chitosan (CT)-modified glassy carbon electrode (GCE) prepared by S. Prakash were used for the detection of As(III) by differential-pulse anodic dissolution voltammetry (DPASV), which has high sensitivity for the detection of As(III) in water due to the unique three-dimensional network and powerful adsorption capacity, and the designed nanostructured electrode has a wide linearity range (10–100 ppb), high sensitivity (0.309 μA ppb^−1^) and a detection limit of 1.20 ppb (16.2 nM) [113]. Riyaz Ahmad Dar used cyclic voltammetry and anodic dissolution voltammetry measurements to evaluate the electrochemical properties of β-cyclodextrin-stabilized AgNPs-GO/GCE for As (III) detection with an approximately threefold increase in peak current compared with GO films alone, showing a wide linear range (13.33–375.19 nM) and high sensitivity (180.5 μA μM^−1^), including a 0.24 nM detection limit [114]. Shao-Hua Wen et al. described the multimodal determination of arsenite (As(III)) in environmental samples by the stimulated response of multi-ligand functionalized silver nanoparticles (Ag NPs) in the electrochemical determination with GSH/DTT/Asn-Ag NPs as the signal probe of the redox electrochemical As(III) sensor. The As(III) concentration increased, the peak currents of Ag NPs in the DPV response curves were recorded, and the calibration curves showed good linearity of peak current intensity with As(III) concentration in the range of 0.01–40 ppb, with detection limits as low as 5.2 ppt [115].

#### 3.1.4. Other Precious-Metal-Modified Electrodes

Agustiany studied the preparation of stable iridium-modified boron-doped diamond electrodes by electrodeposition, and As(III) was detected by cyclic voltammetry [116]. The electrode showed a linear curve in the concentration range of 1–100 μM with a detection limit of 4.64 μM and good stability and reproducibility, with a relative standard deviation of 2.6% under the optimal conditions of a pH 3 phosphate buffer and a scanning speed of 50 mV/s. In addition, the electrode showed good linearity (R^2^ = 0.998) and sensitivity for the measurement of As(III) in both tap water and lake water samples. The test plots are shown in Figure 10. Erfan Mafakheri electrodeposited iridium oxide (IrO_2_) nanotubes in polycarbonate (PC) stencils to form IrO_2_ nanotubes with a uniform diameter of 110 ± 10 nm, an estimated length of 1–3 µm, and IrO_2_ nanotube-modified glassy carbon electrodes for the detection of As(III) [117].

R Gupta et al. deposited metallic ruthenium nanoparticles (RuNPs) on a glassy carbon electrode (GC) with a modified electrode with arsenite selectivity, schematically shown in Figure 11, for the detection of arsenite in water [118]. The differential pulse voltammetry (DPV) based on RuNPs/GC can determine the concentration of arsenite within minutes with a detection limit of 0.1 ppb, a reproducibility of 5.4%, and a sensitivity of 2.38 nA ppb^−1^. RuNPs can be anchored on different chemical platforms such as graphene, electrodes, Fe_3_O_4_, etc. to design robust and reusable electrochemical sensors for the detection of arsenite in various aqueous solutions.

Sthitaprajna Dash investigated the electrodeposition of nanodendritic Pd on poly(3,4-ethylenedioxythiophene) (PEDOT)-modified Pd nanodendritic electrodes for the electroanalysis of As(III) in 1 M HCl solution. A wide detection range of up to 10 μM and a low detection limit of 7 nM (0.52 ppb) can be achieved with a pre-deposition time of 120 s under optimal conditions [119]. Md. Mahbubul Alam electrochemically immobilized Pd nanoparticles on Pt surface in the presence of sodium dodecyl sulfate (SDS) molecules. and its FE-SEM image is shown in Figure 12. The LOD of As(III) was determined to be 0.2 ppb using a Pt-Pd_sds_ sensor [120]. In addition, other noble-metal-modified electrode comparisons are presented in Table 3.

### 3.2. Bimetallic Particle-Modified Electrodes

Precious metal nanomaterials are good for electrode modification, but the cost of modifying electrodes with gold is very high, so people choose to use gold composites for detection, in addition to deriving other bimetallic particles, in order to ensure the detection sensitivity and, at the same time, control the cost of detection. The bimetallic nanomaterials containing gold are Au-Pd, Au-Pt, Au-Ag, Au-Cu, etc. In addition, Fe has an excellent performance in arsenic ion detection, so it has also been studied in Fe-based bimetallic particle-modified electrodes, where bimetallic FePt, FeAu, FePd, and AuPt nanoparticles (NPs) are electrochemically deposited on Si(100) substrates and their electrochemical properties are investigated for As(III) detection. Trace amounts of As(III) can be determined by anodic stripping voltammetry at neutral pH. The synergistic effect with Fe alloying leads to the better performance of Fe precious metal NPs (Au, Pt and Pd) than pristine precious metal NPs (without Fe alloying). Detection limits and linear ranges were obtained for FePt, FeAu, and FePd NPs. The best performance was obtained for FePt NPs with a detection limit of 0.8 ppb and a sensitivity of 0.42 μA ppb^−1^. The selectivity of the sensor was also tested in the presence of large amounts of Cu(II), the most detrimental interfering ion for As detection. Thus, bimetallic NPs are expected to be an effective and high-performance electrochemical sensor for the detection of ultra-trace amounts of arsenic [130]. More electrodes modified with bimetallic materials and their detection performance are summarized in Table 4.

#### 3.2.1. Gold–Platinum Bimetallic Modified Electrodes

A novel and easy-to-use nanohybrid platform suitable for the electrochemical detection of As(III) was prepared based on gold and platinum bimetallic nanoparticles (Au-Pt NPs) and the conducting polymer polyaniline [135]. Good detection limits were obtained by square-wave anodic dissolution voltammetry using modified screen-printed electrodes. The SWASV of 0–15 ppb (As(III) concentration) was recorded (Figure 13a) and this sensor was found to have good linearity in the range of 33–200 nM concentration of As(III) ions, with an LOD up to 19.7 nM, as shown in its calibration plot (Figure 13b).

#### 3.2.2. Gold–Copper Bimetallic Modified Electrode

Recently, a simple hydrothermal method for the preparation of Au and Cu bimetallic nanoparticles of different compositions has been proposed [134]. The electrochemical performance of Au-Cu bimetallic nanoparticles in the determination of As(III) particle concentration was investigated using the square-wave anodic dissolution voltammetry SWASV method, and the SWASV response for As(III) detection in different concentration ranges is shown in Figure 14, where the Cu content in Au-Cu bimetallic nanoparticles is crucial for the detection efficacy. Compared with gold nanoparticles and gold electrodes, Au-Cu bimetallic nanoparticles exhibited better electrochemical performance with a lower detection limit (2.09 ppb) and higher sensitivity (1.63 μA ppb^−1^ cm^−2^). In addition, the Au-Cu bimetallic nanoparticles also exhibited superb anti-interference performance for the detection of As(III).

#### 3.2.3. Silver–Gold Bimetallic Modified Electrode

Reetu Yadav reported a sensor with silver–gold alloy nanoparticles (i.e., silver and gold alloy nanoparticles) modified with glassy carbon electrode and loaded with aptamer [137]. The bimetallic nanoparticles have a large surface area for adhesion with the aptamer, and thus have a large number of binding sites. The detection method uses cyclic voltammetry and differential pulse voltammetry, the electrode is used for the determination of As^3+^ in actual water samples, and the curve is linear when the As^3+^ concentration is 0.01–10 µg/L, with a detection limit of 0.003 × 10^−3^ µg/L. The sensor has good repeatability, stability, and selectivity, and can be applied to the detection of arsenic ion concentration in real water samples.

### 3.3. Other Metals and Their Compound-Modified Electrodes

#### 3.3.1. Fe and Its Compound-Modified Electrodes

In the three-electrode-based electrochemical detection, the adsorption ability of the electrode surface on the target element plays a crucial role in the electrochemical detection performance, and the oxide nanoparticles of Fe become a hot spot for electrochemical detection due to their high adsorption ability and magnetic properties [140]. Pooja Devi et al. reported a chemically reduced rGO/Fe_3_O_4_ nanocomposite-modified glassy carbon electrode, which achieved a low detection limit of 0.12 ppb by square-wave anodic dissolution voltammetry [141]. Haibing Hu’s team prepared Fe_3_O_4_ nanomaterials using a co-precipitation method, resulting in a Fe_3_O_4_ particle size of about 20 nm, which was then compounded with reduced graphene oxide. Under the optimized experimental conditions, the Fe_3_O_4_-rGO-modified glassy carbon electrode showed higher sensitivity (2.15 µA/ppb) for arsenic and achieved low detection limits [142]. The rGO/Fe_3_O_4_ nanocomposite has been shown to be a potential electrochemical and bioelectrochemical sensing material for the simultaneous detection of ascorbic acid, dopamine and uric acid, as well as for the electrocatalytic determination of nitrite. Akajionu Benjamin Chimezie et al. used differential pulsed anodic solvation voltammetry (DPASV) on a screen-printed electrode modified with reduced oxidation. An electrochemical sensor for the determination of As(III) in water resources was developed on the surface of graphene-magnetic nanocomposite (rGO-Fe_3_O_4_/SPEC) using differential pulsed anodic dissolution voltammetry (DPASV) [143]. The schematic diagram is shown in Figure 15. The electrode has a detection limit of 0.1 μg/L for As(III) in drinking water. The sensor has a wide operating range (2–300 μg L^−1^), good repeatability, reproducibility and stability, and is virtually unaffected by common interfering ions.

Hong Cui et al. modified the glassy carbon electrode with a composite of magnetic Fe_3_O_4_ nanoparticles and gold nanoparticles, and the detection limit was 0.00097 ppb under optimal conditions [59]. Chao Gao et al. proposed an Fe_3_O_4_-RTIL (room temperature ionic liquid) composite-modified screen-printed carbon electrode (SPCE) in order to achieve ultra-low detection limits while reducing the dependence on precious metal gold, and obtained an ultra-low detection limit of 8 × 10^−4^ ppb by square-wave anodic solvation voltammetry (SWASV) while achieving a high sensitivity of 4.91 μA ppb^−1^ [144]. Other ferrites also have great potential for electrochemical detection. Shaofeng Zhou et al. reported Au nanoparticle-decorated mesoporous MnFe_2_O_4_ nanocrystal clusters for the detection of As(III) in water samples by square-wave anodic dissolution voltammetry (SWASV) with good reproducibility, stability and reproducibility, as well as interference resistance [74]. In a recent study, Hong-QiHuang et al. proposed a successful electrochemical sensor driven by noble metal-free layered porous Fe_3_O_4_/Co_3_S_4_ nanosheets for As(III) analysis, and obtained a considerable sensitivity of 4.359 μA/ppb for the electrochemical detection of As(III) in 0.1 M HAc-NaAc (pH 6.0) by square-wave anodic dissolution voltammetry (SWASV). The improved electrochemical performance of As(III) is attributed to its nanoporous structure, the presence of oxygen vacancies and the strong synergistic coupling effect between Fe_3_O_4_ and Co_3_S_4_ species [145].

#### 3.3.2. Manganese and Cerium Oxide-Modified Electrodes

Due to the synergistic effect, Mn_2_O_3_/CeO_2_ nanocubes have a high adsorption capacity for As(III), so its detection sensitivity is higher than any kind of oxide. Combined with the sensing properties of gold (Au) for As(III), a sensing material based on Mn_2_O_3_/CeO_2_ nanocubes modified with gold electrode was fabricated [146], as shown in Figure 16. Under the optimized conditions, the sensitivity of the sensor was 0.0414 mA ppb^−1^ and the limit of detection (LOD) was 3.35 ppb with good stability and reproducibility, and the electrode had good selectivity for the presence of common interfering ions.

Manganese oxide (MnO_2_) can be used as an active electrode material due to its good redox properties, porosity, low cost, and large specific surface area. In addition, the electrocatalytic properties of the composites can be further improved through synergistic effects by immobilizing metal oxides on the surface of polyhydroxytyramine (POT) and graphene oxide (rGO) composites. Sathish Kumar Ponnaiah et al. used a novel manganese dioxide/polyhydroxytyramine/reduced graphene oxide nanocomposite (MnO_2_/POT/rGO/GCE) to fabricate sensing electrodes with a wide linear range (0.01–0.900 ppb) and minimum detection limit (42.0 ppt), and excellent selectivity, stability and reproducibility [147]. The process schematic is shown in Figure 17.

#### 3.3.3. Cobalt Oxide-Modified Electrodes

An electrochemical sensor based on cobalt oxide nanoparticles was developed by Abdollah Salimi et al. Cobalt oxide (CoOx) nanoparticles were prepared from an aqueous buffer solution of CoCl_2_ using cyclic voltammetry and deposited on the surface of a glassy carbon electrode. Then, As(III) was detected by cyclic voltammetry, and a detection limit of 11 nM was achieved. The authors concluded that immobilizing cobalt oxide nanoparticles on the surface of GC electrode seems to be an efficient method to develop a new class of sensitive, stable and reproducible electrochemical sensors for As(III) [148]. Chun-Yang Li combined the excellent catalytic properties of AuNPs with the high adsorption capacity of Co_3_O_4_ nanomaterials to construct an ultra-sensitive electrochemical sensor for electrochemical analysis of As(III) by homogeneously assembling gold nanoparticles on porous cobalt oxide (Co_3_O_4_) microsheets to form nanocomposites [98]. The experimental results show that the AuNPs/ Co_3_O_4_ nanocomposite-modified SPCE achieved an ultra-high sensitivity of 12.1 ± 0.2 μA ppb^−1^ and a detection limit of 0.09 ppb for As(III) using the SWASV method. This excellent electrochemical performance was attributed to the high adsorption capacity of the porous Co_3_O_4_ microporous sheet and AuNPs for the favorable electrocatalysis of As(III) reduction. In addition, the method also exhibits good anti-interference performance in the presence of other metal ions (Cu(II), Pb(II), Cd(II), etc.) with good stability and reproducibility. Most importantly, the electrochemical sensor has been successfully applied to the electroanalysis of As(III) in water and human serum samples, which provides a new approach to design sensitive and stable electrochemical sensors. The schematic diagram of the electrochemical analysis is shown in Figure 18 below.

#### 3.3.4. Tin Oxide-Modified Electrodes

Tin oxide (SnO_2_) has become the material of choice for arsenic (As^3+^) redox sensing due to its high catalytic activity, environmental performance, wide band gap (3.64 eV) and high specific surface area (nanoscale) [149]. Tian-JiaJiang et al. reported an ultrathin SnO_2_ nanosheet for the modification of gold electrodes, resulting in enhanced adsorption capacity on the gold electrode surface [150]. Gaurav Bhanjana et al. synthesized SnO_2_ nanopins (particle size 60–80 nm) by the chemical precipitation method, characterized their elemental, topological, morphological and structural features, and then coated these nanopins on the surface of pencil cores (containing graphite/carbon(C)) to serve as working electrodes to prepare nanomaterial sensors for the detection of arsenic ions [151]. By the electrochemical determination of arsenic in real samples, the sensor has a detection limit of 10 ppb, a linear range of 50–500 ppb, and a sensitivity of 28.13 μA ppb^−1^ cm^−2^. The experimental results provide a feasible method for the field detection of As^3+^ in environmental samples such as food, beverages, industrial samples, and wastewater. The modified electrode current response relationship under certain conditions is shown in Figure 19 and Figure 20.

#### 3.3.5. Strontium Compound-Modified Electrodes

A. Karthika et al. reported a glassy carbon electrode modified with strontium titanium trioxide (SrTiO_3_) and β-cyclodextrin (β-CD)-based nanocomposite, for the determination of toxic As(III) ions in water and serum samples [152]. The prepared SrTiO_3_/β-CD nanocomposite-modified glassy carbon electrode has a high specific surface area and a sensitive electrochemical response. After testing, the oxidation peak current of As(III) increased linearly with the concentration in the concentration range of 10–140 μM of As(III) particles, and the detection limit was 0.02 μM. The electrode is stable, sensitive and reproducible for the detection of As(III) in water and serum.

#### 3.3.6. Bismuth Compound-Modified Electrode

Thabile Ndlovu electrodeposited a bismuth film onto an exfoliated graphite (EG) electrode at a potential of −600 mV. The modification of EG resulted in an increase in the electroactive surface area of the electrode, and square-wave anodic dissolution voltammetry using the modified electrode (EG-Bi) in As(III) solution was able to detect 5 ppb of As(III) and was insensitive to many interfering cations except Cu(II) [153]. Potlako J. Mafa also electrodeposited bismuth nanoparticles onto graphite electrodes and used square-wave anodic solvation voltammetry (SWASV) to co-detect heavy metal ions in water samples with a detection limit of 0.014 ppb for As(III) under optimized experimental conditions [154]. In addition to bismuth nanoparticles, Lignesh Durai reported a novel and facile hydrothermal synthesis of bismuth vanadate (BiVO_4_) nanoflakes for the trace detection of arsenic in biological samples by the electrodeposition of a screen-printed carbon electrode (SPCE) coated with polyaniline (PANI), with the As(III) sensing mechanism as shown in Figure 21. The sensor can detect As^3+^ ions by the differential pulse dissolution voltammetry (DPASV) technique with a significantly low limit of detection (LOD) of 0.0072 ppb and a sensitivity of 6.06 μA ppb^−1^ cm^−2^ with a linear range of 0.01–300 ppb [155].

#### 3.3.7. Zirconium Compound-Modified Electrodes

Gaurav Bhanjana used gold electrodes modified with zirconia nanocubes synthesized by a facile hydrothermal route, and electrochemical sensing of arsenic was achieved by cyclic voltammetry (CV) and chronoamperometry with a sensitivity of 550 nA cm^−2^ ppb^−1^ and a detection limit of 5 ppb (linear range of 5–60 ppb, response time below 2 s). The synthesized nanoparticles are nanocubes, and from the CV plots under different conditions (Figure 22) it can be seen that the peak oxidation current is more pronounced for zirconia nanocube-modified electrodes than for zirconia nanoparticle-modified electrodes, which are used as effective electrocatalysts in the direct redox sensing of arsenic [156]. In addition, zirconia composites were used for the electrochemical detection of arsenic. MengYang used AuNPs/CeO_2_-ZrO_2_ nanocomposite-modified glassy carbon electrodes (GCE) to fabricate a sensing interface for the sensitive and accurate analysis of As(III) in near groundwater pH values, and square-wave anodic dissolution voltammetry (SWASV) was used to determine As(III) in real water samples. Thanks to the strong adsorption capacity of CeO_2_-ZrO_2_, the electroanalytical sensitivity and theoretical detection limit of As(III) were 0.976 μA ppb^−1^ and 0.137 ppb, respectively, at the optimal parameters. In addition, the method has good anti-interference performance [85]. The performance analysis of different electrodes modified with metals other than noble metals and their compounds for the electrochemical detection of arsenic is given in Table 5.

### 3.4. Carbon Nanomaterial-Modified Electrodes

In electrochemical sensing, carbon-based electrodes such as glassy carbon electrodes, screen-printed carbon electrodes and graphite–carbon paste electrodes are widely used for arsenic ion detection. In addition, due to the unique electronic properties of carbon-based nanomaterials, carbon nanomaterials have proven to be very suitable for the modification of working electrodes in the electrochemical detection of arsenic. Carbon nanomaterials include single-walled carbon nanotubes (SWNT), multi-walled carbon nanotubes (MWNT), graphene, nanodiamond, fullerene, and graphene quantum dots. The most widely studied carbon nano-derivatives in electrochemical sensing are carbon nanotubes and graphene [180]. The performance analysis of different carbon nanomaterials and their composite-modified electrodes for the electrochemical detection of arsenic is statistically presented in Table 6.

#### 3.4.1. Carbon Nanotube-Based Detection of Arsenic Ions

Carbon nanotubes (CNT) are an excellent support structure due to their large effective detection surface, fast electron transfer rate compared with bulk carbon electrodes, high electrocatalytic activity and low electrode contamination, which can further help to improve electrochemical analysis performance by immobilizing other chemical species, such as metal NPs and organic molecules [14]. The main features of carbon nanotubes in electrochemical sensors are a fast response and low detection limits.

He Xu et al. chose to form a composite with carbon nanotubes for the electrochemical detection of arsenic using Pt nanoparticles, as shown by the TEM image of the composite (Figure 23), where Pt nanoparticles are clearly decorated on carbon nanotubes, reflecting a higher electroactive area than the Pt nanoparticle modification alone [104]. Yaxiong Liu et al. proposed a single-layer low-resistance single-walled carbon nanotube-modified glassy carbon electrode for the electrochemical detection of arsenic with non-covalent SH groups sensitive to As(III), achieving an ultra-low detection limit of 0.008 ppb [188]. In addition to compounding with metal particles, carbon nanotubes can also be combined with biomolecules and with DNA. Yaxiong Liu’s team developed a layer-by-layer assembly of DNA-functionalized single-walled carbon nanotubes that achieves a detection limit of 0.05 ppb in a near-physiological environment and can be reused multiple times [189]. Subramanian Nellaiappan et al., on the other hand, proposed a gold nanoparticle/carbon nanofiber/chitosan chemically modified carbon screen-printed electrode by simultaneously combining carbon nanofibers with metal particle biomolecules, achieving comparable results to inductively coupled plasma-emission spectroscopy [83].

#### 3.4.2. Graphene-Based Detection of Arsenic Ions

Graphene is a new material in which carbon atoms are tightly packed into a single two-dimensional honeycomb lattice structure. This material has excellent properties, with high strength and good toughness, and good electrical conductivity, and can be used directly as a modification material for electrodes. In addition, graphene has a characteristic that is very favorable for loading other materials, which is that it has many smooth folds, so there will be many graphene and other material composites to modify the electrode, so that the characteristics of the loading material can be more obvious.

Haibing Hu’s team performed the electrochemical characterization of bare glassy carbon electrode, rGO, Fe_3_O_4_, and Fe_3_O_4_-rGO nanocomposite-modified glassy carbon electrode in a specific detection solution in the preparation of Fe_3_O_4_-rGO nanocomposite, and found that the electrochemical performance of glassy carbon electrode modified by rGO or Fe_3_O_4_ only was not as good as that of bare glassy carbon electrode, although it was better than that of bare glassy carbon electrode, while the electrochemical performance of glassy carbon electrode modified by both composites was very good, which indicated that the large surface area provided by rGO caused the Fe_3_O_4_ particles to adhere well to the surface [142]. In addition to metal oxides, graphene can also form composites with metals [185], multi-walled carbon nanotubes [195], precious metals [114], and biomolecules [197], among others.

Carbon nanoparticles and other carbon-based nanomaterials have been used for signal enhancement in electrochemical sensors and biosensors due to their advantageous specific surface area. An electrochemical sensor based on carbon nanoparticles (CNPs) and gold nanoparticles (AuNPs) comprising an immobilized platform for As(III) detection in water was reported in 2019 [136]. The carbon–gold nanoplatform was prepared by drop coating CNPs on a glassy carbon electrode (GCE), followed by the electrodeposition of AuNPs on the CNPs-modified electrode under certain conditions. The sensor has a detection limit of 0.092 ppb and exhibits insensitivity to the interference of Cd^2+^, Cu^2+^, and Hg^2+^, providing an interference reduction method for the electrochemical detection of arsenic.

### 3.5. Biomolecule-Modified Electrodes

#### 3.5.1. Arsenic Detection Based on DNA-Modified Electrodes

DNA in DNA-based biosensors provides biologically recognizable components with three modes of interaction [198], namely electrostatic interactions with negatively charged phosphates, binding interactions with minor and major grooves of the DNA double helix, and embedding between natural DNA stacked base pairs.

An advanced DNA biosensor was reported by J. Labuda et al. [199]. Using the Co(III) complex with 1,10-phenanthroline, [Co(phen)_3_]^3+^, as an electrochemical DNA marker and the Ru(II)complex with bipyridyne, [Ru(bipy)_3_]^2+^, as a DNA oxidation catalyst, calf thymus DNA (CT-DNA) immobilized on the surface of a screen-printed electrode (SPE) was placed in aqueous solutions of different concentrations of As(III), As(V), dimethylarsenic acid, phenylarsenic and p-arsenic acid. Although this system was reported to have a poor detection limit (75 mg/L), it showed a successful correlation between DNA-labeling signals and As(III) levels. Liu and Wei exploited the high electrical conductivity of carbon nanotubes (CNTs) to construct electrochemical biosensors and explored the concept of the direct oxidation of As(0) to As(III) on DNA-functionalized single-walled CNT-modified glassy carbon electrodes [189]. The developed biosensor was operationally stable over a wide pH range with a detection limit (S/N = 3) of 0.05 μg L^−1^ at pH 7.0. and demonstrated the ability to be reused 16 times. Shaohua Wen described a voltammetric method for the determination of arsenite (As(III)) based on the specific binding of As(III) to probe DNA (SBP DNA; single-stranded DNA) and the electrochemical indicator methylene blue (MB), the fabrication of which is schematically shown in Figure 24. Upon addition of As(III), it specifically binds to SBP DNA, which leads to conformational changes and the dissociation of SBP DNA from the electrode into the solution. As a result, the amount of MB remaining on the modified electrode is reduced, which decreases the peak MB current. Under optimized conditions, As(III) was quantified by measuring the DPV response of MB absorbed by the SBP/CP hybrid at the electrode surface, and the reduction peak current was linearly related to the logarithmic value of As(III) concentration, yielding a linear concentration range of 0.1–200 ppb and a detection limit as low as 75 ppt [200].

#### 3.5.2. Aptamer Sensors for Arsenic Detection

An aptamer-based biosensor is a small device that assembles one or more biomaterials/nanomaterials onto an electrode transducer, and electrochemical impedance spectroscopy (EIS), differential pulse voltammetry (DPV), etc., are often used with aptamer sensors for arsenic detection [201]. For example, Baghbaderani and Noorbakhsh constructed several aptamer sensors based on electrochemical signals for the determination of As(III) [202]. They designed an unlabeled impedance aptamer sensor for highly sensitive As(III) determination using a chitosan-Nafion(Chit-Naf) compound as an excellent conductive surface platform and a novel carbon nanotube based on the signal amplification process. The EIS experimental results show that the glassy carbon electrode (GCE) modified by Chit-Naf has higher electron transfer kinetics compared with bare GCE, GCE/Naf, and GCE/Chit electrodes, which provides great feasibility for an effective platform for biosensor design. In this work, based on a carbon nanotube–bovine serum albumin (CNT–BSA) hybrid system, they also used a signal amplification process to achieve an LOD of 74 pM. Lin Cui et al. designed an electrochemical aptamer sensor for the detection of As(III) based on gold-nanocoated screen-printed carbon electrodes (AuNPs/SPCE) [203], and the detection schematic is shown in Figure 25. By immobilizing the Ars-3 aptamer on AuNPs/SPCE, the Ars-3 aptamer is able to adsorb cations through electrostatic interactions with polydiallyldimethylammonium chloride (PDDA) and repel other cations. In the presence of arsenite, the Ars-3 conformation changes due to the formation of Ars-3/As(III) complexes, which reduces the adsorption of more positively charged electrochemically active indicator [Ru(NH_3_)_6_]^3+^ on the surface of the PDDA adsorption electrode as a means to achieve coupling, thus enabling detection.

#### 3.5.3. Arsenic Detection Based on Other Biomolecules

In addition to DNA with aptamers, certain proteins have been used as materials for the electrochemical detection of arsenic, and most protein-based arsenic detection is based on the inhibition phenomenon. Cytochrome-C (Cyt-C), an important component of the mitochondrial electron transport chain, is sensitive to all toxic compounds and is also used as a biorecognition element [204]. An electrochemical biosensor was constructed using Cyt-C, immobilized on a boron-doped diamond electrode. Square-wave voltammetry (SWV) and electrochemical impedance spectroscopy (EIS) were performed to investigate the interaction of Cyt-C with arsenic and cyanide. Subtractive normalized Fourier transform infrared spectroscopy (SNFTIR) was performed to confirm the effective protein adsorption onto the electrode. UV–vis studies of Cyt-C with the analytes confirmed the correct binding. The results indicate that their interaction was through the amino acids of the basic protein structure rather than through the heme portion of Cyt-C. Jae-Hoon Hwang et al. developed a novel As(III) sensor by depositing iron–chitosan complexes on screen-printed carbon electrodes using electrodeposition [205]. Mine wastewater and soil leachate were tested by square-wave anodic dissolution voltammetry. The detection limits of the Fe–chitosan-coated electrode were 1.12 ppb for mine wastewater and 1.01 ppb for soil leachate, both of which were significantly lower than the WHO requirements. The interference of Cu^2+^ ions had little effect on the detection, indicating that the chitosan-coated iron carbon could improve the stability. The sensor has high sensitivity and selectivity and provides a reliable level of detection of As(III) concentration in leachate from actual wastewater and contaminated sites. Suparna Saha et al. modified glassy carbon electrode with chitosan–Fe(OH)_3_ composite and reducing agent L-cysteine [158], and its detection schematic is shown in Figure 26, under optimal optimized conditions, by differential pulse. The anodic dissolution voltammetry achieved a detection limit of 0.072 ppb in the linear interval of 2–100 ppb and avoided the interference of common co-existing ions. More examples of biomolecule-modified electrodes for the detection of arsenic are given in Table 7.

### 3.6. Others

#### 3.6.1. Silicon and Its Compound-Modified Electrodes

Suhainie Ismail et al. developed an efficient electrochemical detection method for arsenite using linear scanning anodic solvation voltammetry (LSASV) based on silicon nanoparticles and gold nanoparticles (SiNPs/AuNPs/SPCE) modifying the screen-printed electrode surface [93]. The electrode showed good linearity in the concentration range of 10–100 ppb with a detection limit of 5.6 ppb. Multiple co-existing ions—Pb^2+^, Ni^2+^, Zn^2+^, Hg^2+^ and Cu^2+^—in the water samples did not interfere with the detection of arsenite. The method is highly sensitive and reproducible with a relative standard deviation of 4.52%, which is promising for application. In addition, they also tried to detect As(III) using silicon nanoparticles (SiNPs)-modified screen-printed electrodes (SPCE) and tested the electrochemical response of the electrode to arsenic using cyclic voltammetry (CV) and linear scanning anodic solvation voltammetry (LSASV) [219]. Under the optimized conditions, the peak anode current showed good linearity in the concentration range of 5–30 μg/L As(III) with a detection limit of 6.2 μg/L. This method can effectively detect As (III) in real water samples with low fabrication cost, good reproducibility and stability. In addition to silicon nanoparticles, Feng Sun prepared nano-Au/SiO_2_ modified GCE by a one-step method. This nanohybrid material was used for the electrochemical detection of As (III). The calculated LOD was 0.07 μg/L, with a linear detection range of 0.1–40 μg/L [80].

#### 3.6.2. Novel Polymer-Modified Electrodes

Mohammed M. Rahman synthesized a new class of thermally stable hybrid poly(arylene)(azomethanes) and copoly(arylene)(azomethanes) (PAAP) based on diarylidenecycloalkanes by solution polycondensation, combined with a conductive nafion (5%) coating agent-modified glassy carbon electrode for the detection of arsenic by the I-V method, which exhibits higher sensitivity and selectivity for As^3+^ ions. Based on the calibration curve, the sensitivity and detection limits were calculated as 2.714 μA μM^−1^ cm^−2^ and 6.8 ± 0.1 nM, respectively, and this novel method provides a new route for the electrochemical detection of arsenic ions [220]. Wuwei Ma et al. proposed an electrochemical sensor based on ion-imprinted polymers (IIPs) and nanoporous gold (NPG)-modified gold electrode (IIP/NPG/GE) for the determination of arsenic ions (As^3+^) in different kinds of water, which was prepared by the electrodeposition of nanoporous gold on the gold electrode, and then a layer of IIPs with As^3+^ as the template ion was synthesized in situ on the NPG surface by electropolymerization. The IIPs/NPG/GE formation process is shown in Figure 27. The linear range of As^3+^ was obtained from 2.0 × 10^−11^ to 9.0 × 10^−9^ M by cyclic voltammetry, and the lower limit of detection was 7.1 × 10^−12^ M after the calibration curve [221].

## 4. Conclusions

Arsenic contamination has seriously endangered the living environment and health of human beings, and achieving the efficient and reliable measurement of arsenic ions has gradually become a popular research area in the scientific community. The electrochemical detection method based on a nanomaterial-modified electrode has become a mainstream analytical method for measuring inorganic arsenic, with many advantages, such as simple operation, high sensitivity, good selectivity, low cost, and rapid portability. The working electrode is modified by nanomaterials so as to improve the performance of the electrochemical sensor, as described in this paper. The use of noble metal materials to modify the electrode can increase the mass transfer and reduce the effect of solution resistance; due to the synergistic effect, the bimetallic materials can ensure the detection performance while controlling the detection cost. Although noble metals show excellent performance in the electrochemical detection of arsenic, no noble metal nanomaterials are considered to be mainstream for the electrochemical detection of arsenic. The use of other metals and their compounds can achieve low cost, high sensitivity, and strong interference resistance; secondly, biomolecule-based electrochemical sensors for arsenic have better reproducibility and feasibility, and are increasingly being used in clinical diagnosis, food analysis, and environmental monitoring.

Most electrochemical studies are conducted under acidic conditions (i.e., acidic buffer solutions are used as solvents in sample preparation, along with the more common buffer solutions such as phosphate, acetic acid, hydrochloric acid, nitric acid buffers, etc.). Precious metal nanoparticles, bimetallic nanomaterials, metal oxide nanomaterials, and other modified electrodes show better performance and obtain better sensitivity in acidic environments. However, detection under ambient pH conditions has several advantages, such as avoiding unexpected changes in As morphology during acidification and simplifying the experimental procedures during field detection. In the operation of electrode preparation, deposition techniques, including electrodeposition and chemical deposition, are mostly used for metal nanoparticles, and drop casting is less frequently used. For metal oxide nanoparticles, the drop-casting method is mostly used directly. In real water samples, there may be interference from co-existing ions, such as Pb(II), Cu(II), Ni(II), Co(II), Cr(III), Zn(II), and NO_3_^−^. The electrodes in most of the previous studies showed high anti-interference and selectivity, and also performed well when the concentration of interfering ions was much higher than that of arsenic.

A method suitable for field analysis which can achieve a low detection limit (within 10 ppb) is urgently needed to detect arsenic in drinking water, and the use of a nanomaterial-modified electrode electrochemical system for this purpose represents a great opportunity. Research on new nanomaterials continues to make progress, such as reduced graphene oxide (rGO) and other metal oxide composites, which have been shown to have good detection performance, showing that metals and their compound nanomaterials for the detection of arsenic ions in water have a bright future.

## Figures and Tables

**Figure 1 nanomaterials-12-00781-f001:**
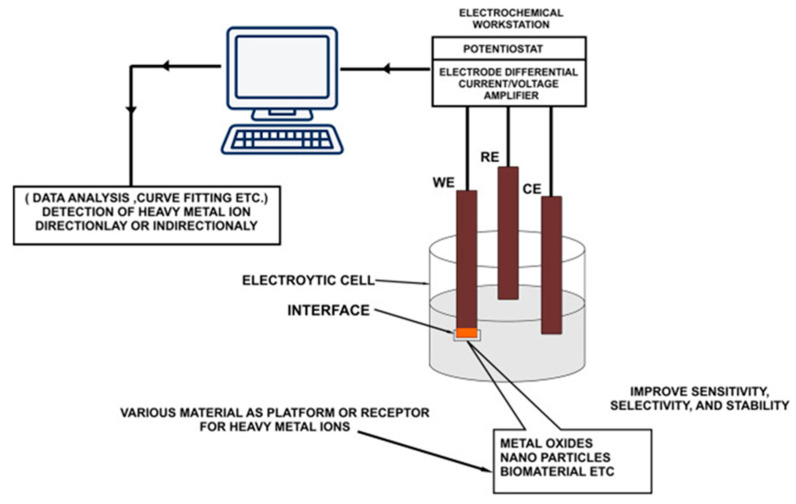
General apparatus for electrochemical detection of arsenic. (Reprinted with permission from [13], Copyright 2017, Elsevier B.V.)

**Figure 2 nanomaterials-12-00781-f002:**
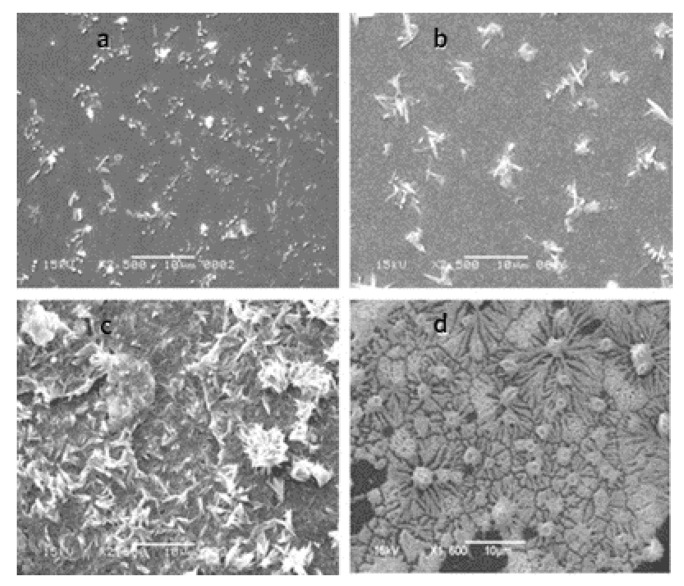
SEM images of Ibu-NSs at (**a**) 1, (**b**) 10, (**c**) 20, and (**d**) 30 min of heating time. (Reprinted with permission from [46], Copyright 2012, Elsevier B.V.)

**Figure 3 nanomaterials-12-00781-f003:**
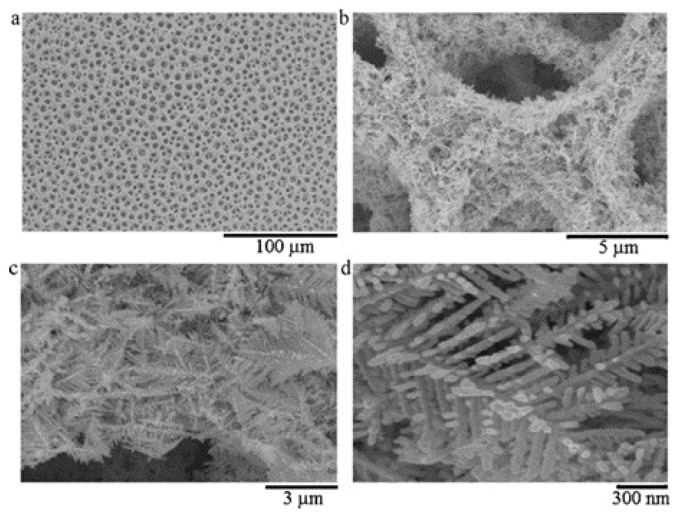
SEM images of porous structures of 3D Au nanodendritic networks with four different fields ((**a**) 100 μm, (**b**) 5 μm, (**c**) 3 μm, and (**d**) 300 nm) of view. (Reprinted with permission from [47], Copyright 2011, Elsevier B.V.)

**Figure 4 nanomaterials-12-00781-f004:**
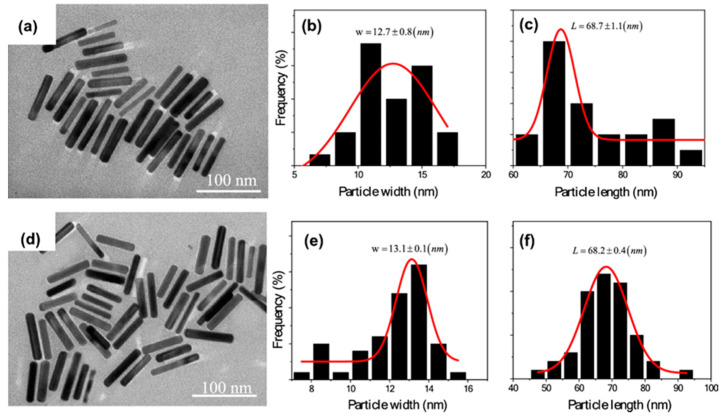
TEM images, diameter, and length distribution of GNR-CTAB (**a**–**c**) and GNR-PSS (**d**–**f**). (Reprinted with permission from [48], Copyright 2022, Springer Nature Switzerland AG.)

**Figure 5 nanomaterials-12-00781-f005:**
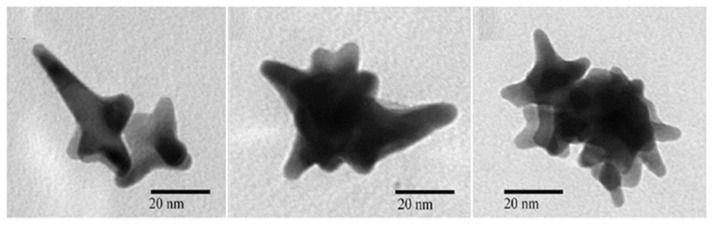
Representative TEM images of AuNS. (Reprinted with permission from [49], Copyright 2022, Springer Nature Switzerland AG.)

**Figure 6 nanomaterials-12-00781-f006:**
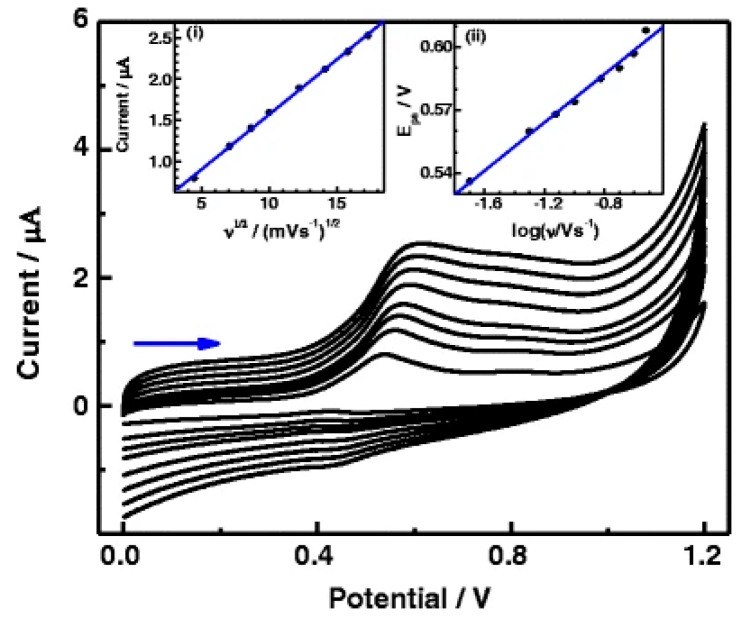
Cyclic voltammetric response of GC/Au-bt to 100.0 μM As(III) oxidation at different scan rates. Inset (i) shows the plot of the oxidation peak currents against the square root of scan rates and inset (ii) shows the plot of the oxidation peak potential against the log of scan rate. (Reprinted with permission from [53], Copyright 2016, Springer Nature).

**Figure 7 nanomaterials-12-00781-f007:**
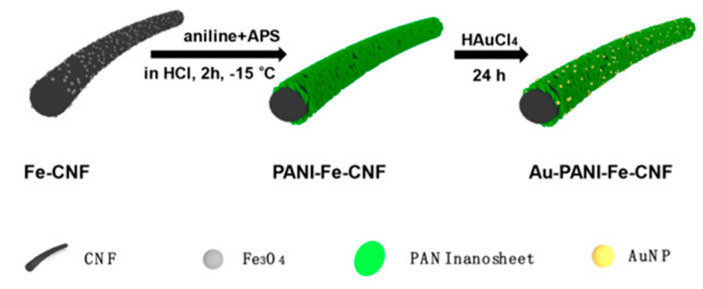
Schematic diagram of the preparation process of Au-PANI-Fe-CNFs. (Reprinted with permission from [54], Copyright 2020, Elsevier B.V.)

**Figure 8 nanomaterials-12-00781-f008:**
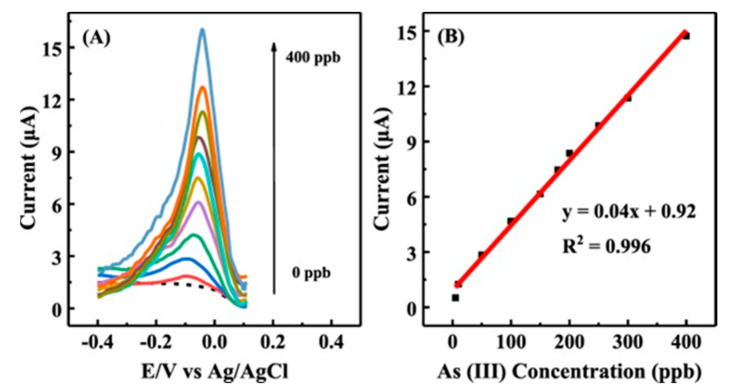
(**A**) SWASV response of different concentrations of As(III) on Au-PANI-Fe-CNFs/GCE. (**B**) Linear calibration curve of peak current versus As(III) concentration (0–400 ppb). (Reprinted with permission from [54], Copyright 2020, Elsevier B.V.)

**Figure 9 nanomaterials-12-00781-f009:**
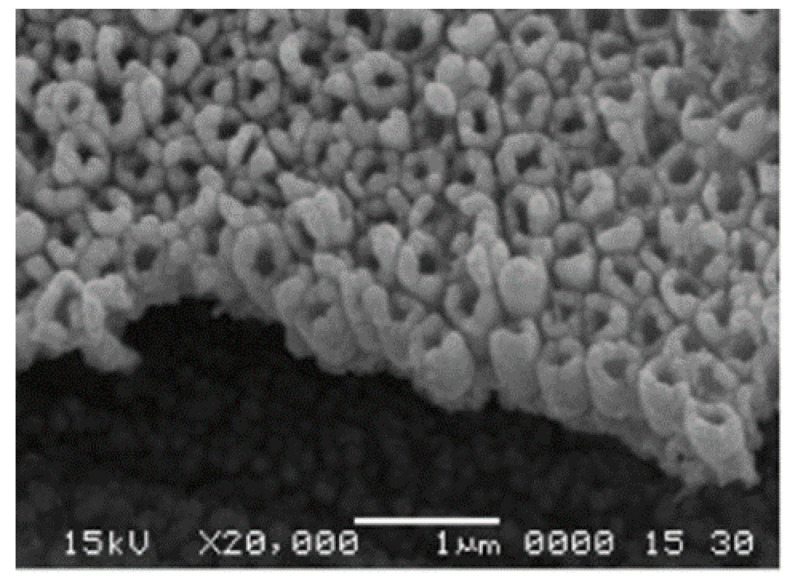
SEM image of cross-sectional view of PtNTAEs. (Reprinted with permission from [103], Copyright 2008, Elsevier B.V.)

**Figure 10 nanomaterials-12-00781-f010:**
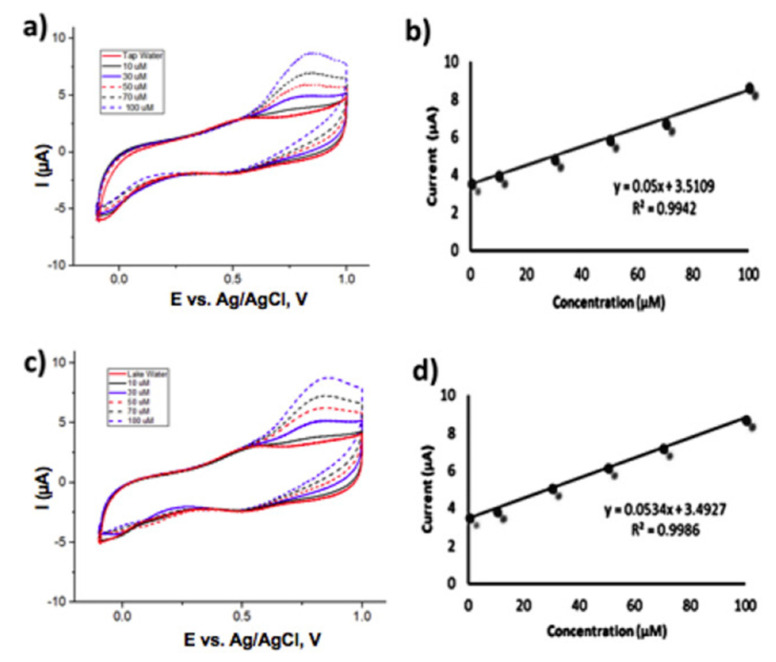
Cyclic voltammetry responses of different spike concentrations of arsenic (III) in (**a**) tap water and (**c**) lake water samples; scan rate 50 mV/s at Ir-BDD prepared using complete step deposition. (**b**) and (**d**) depict the dependence of current responses on arsenic (III) concentrations. (Reprinted with permission from [116], Copyright 2020, Elsevier B.V.)

**Figure 11 nanomaterials-12-00781-f011:**
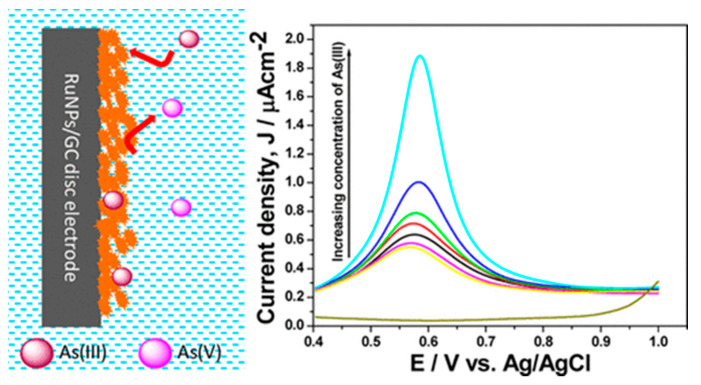
Schematic diagram of RuNPs/GC-based As electrochemical detection. (Reprinted with permission from [118], Copyright 2016, American Chemical Society.)

**Figure 12 nanomaterials-12-00781-f012:**
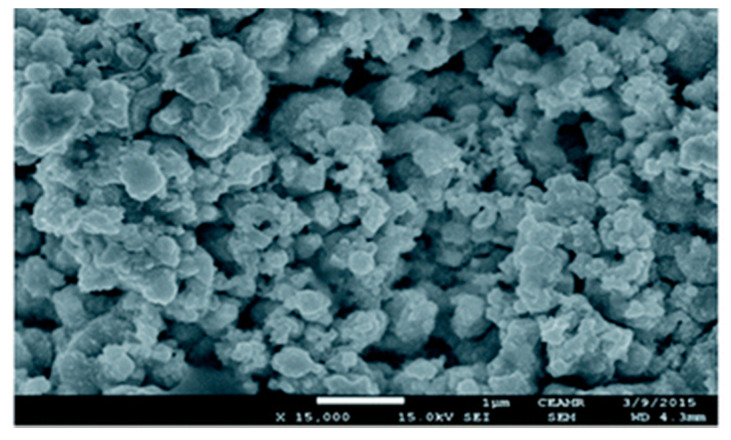
FE-SEM image of Pd particles deposited on Pt surface. (Reprinted with permission from [120], Copyright 2018, The Royal Society of Chemistry.)

**Figure 13 nanomaterials-12-00781-f013:**
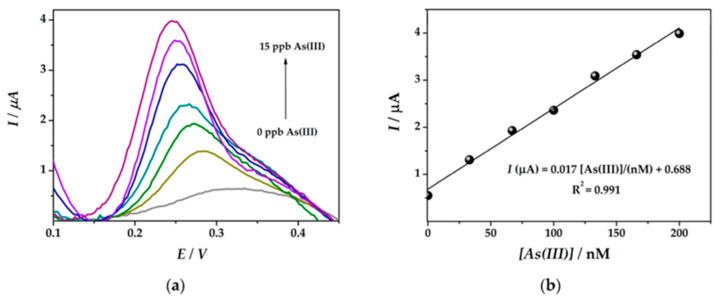
(**a**) SWASV curves obtained at the Au-PtNPs/SPCE platform. (**b**) Corresponding calibration plots (Reprinted with permission from [135]).

**Figure 14 nanomaterials-12-00781-f014:**
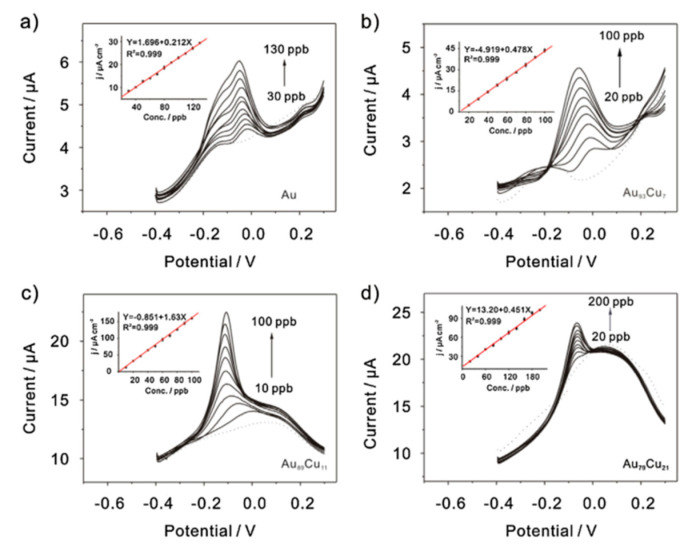
SWASV response of (**a**) Au, (**b**) Au_93_Cu_7_, (**c**) Au_89_Cu_11_, and (**d**) Au_79_Cu_21_ bimetallic nanoparticle-modified GCE for As(III) detection at different concentration ranges. (Reprinted with permission from [134], Copyright 2016, Elsevier B.V.)

**Figure 15 nanomaterials-12-00781-f015:**
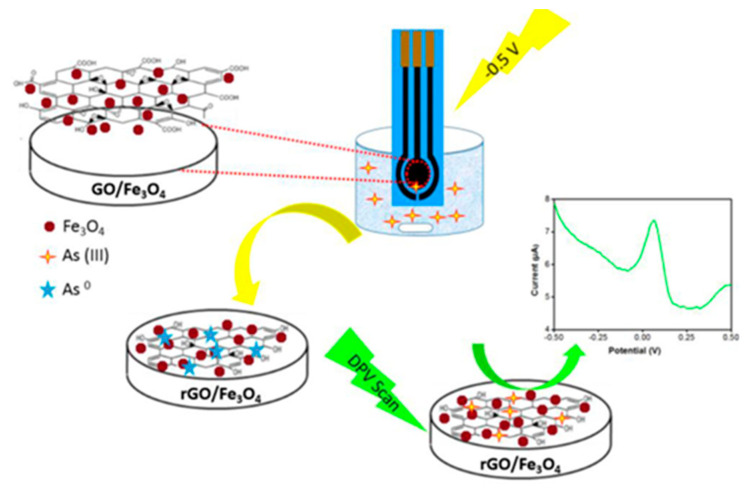
Schematic diagram of rGO-Fe_3_O_4_/SPEC-based electrochemical analysis of arsenic ions. (Reprinted with permission from [143], Copyright 2017, Elsevier B.V.)

**Figure 16 nanomaterials-12-00781-f016:**
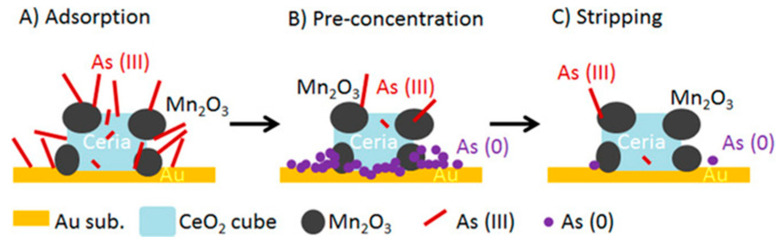
Schematic diagram of electrochemical analysis of arsenic on Mn_2_O_3_/CeO_2_/gold electrode. (**A**) Adsorption, (**B**) pre-concentration, (**C**) stripping. (Reprinted with permission from [146], Copyright 2018, Wiley-VCH Verlag GmbH & Co. KGaA, Weinheim.)

**Figure 17 nanomaterials-12-00781-f017:**
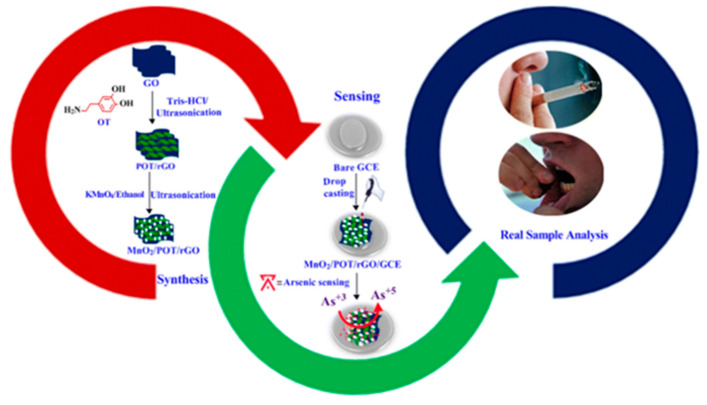
Synthesis of MnO_2_/POT/rGO nanocomposites and arsenic detection process. (Reprinted with permission from [147], Copyright 2020, Springer Nature Switzerland AG.)

**Figure 18 nanomaterials-12-00781-f018:**
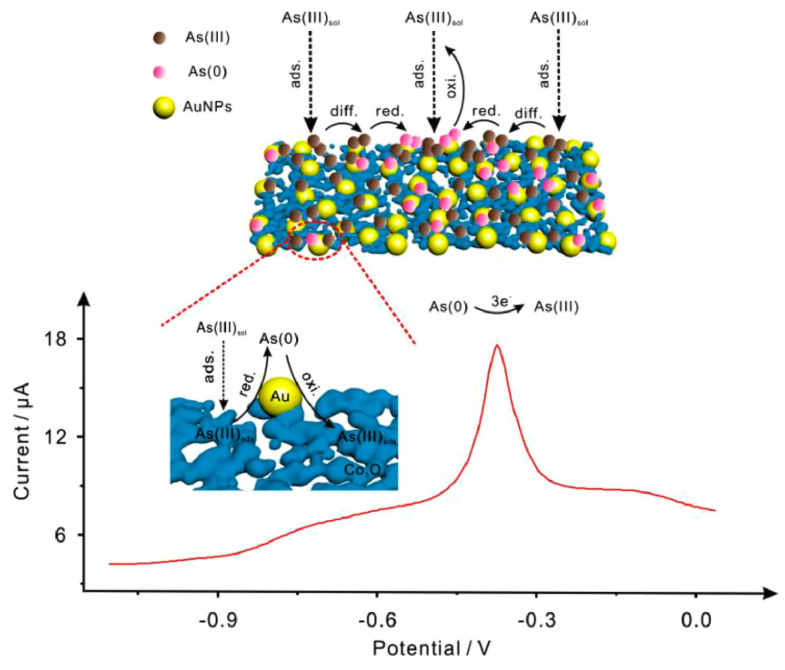
Electrochemical analysis of arsenic ions in solution based on AuNPs/Co_3_O_4_/SPCE. (Reprinted with permission from [98], Copyright 2020, Elsevier Ltd.)

**Figure 19 nanomaterials-12-00781-f019:**
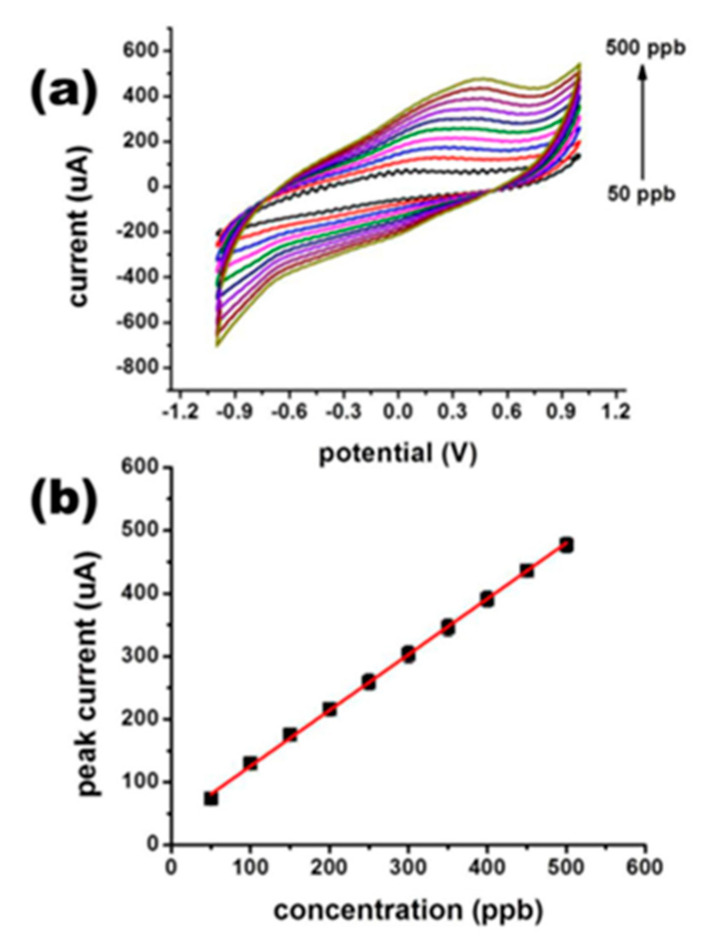
(**a**) Current response to different concentrations of arsenic; (**b**) oxidation peak current and concentration linear calibration curve. (Reprinted with permission from [151], Copyright 2018, Elsevier B.V.)

**Figure 20 nanomaterials-12-00781-f020:**
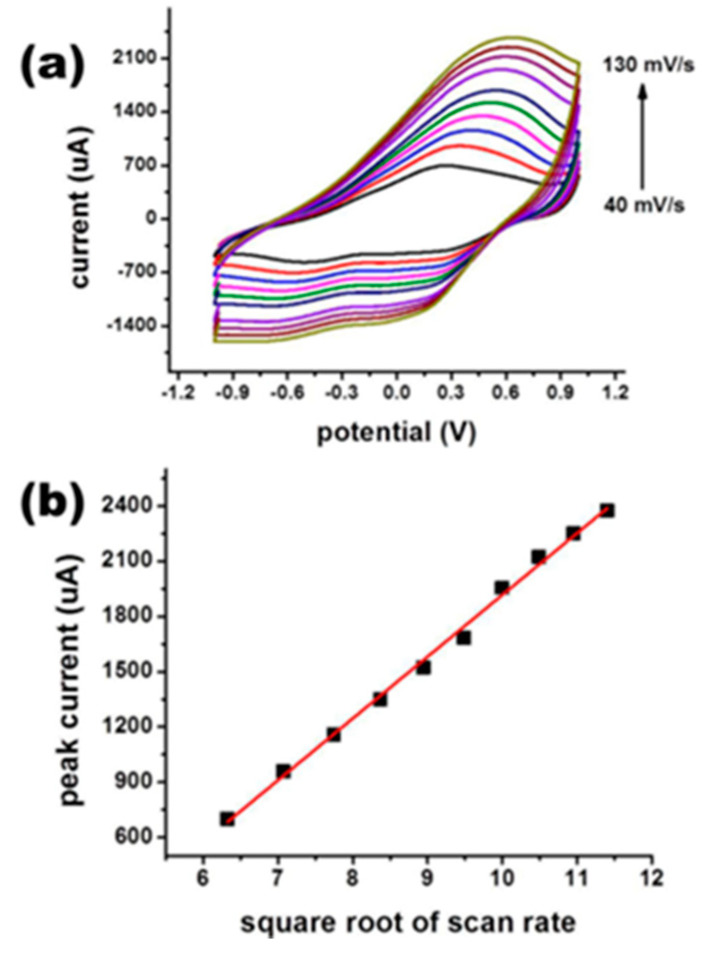
(**a**) Current response at different scan rates; (**b**) oxidation peak current and scan rate linear calibration curve (Reprinted with permission from [151], Copyright 2018, Elsevier B.V.)

**Figure 21 nanomaterials-12-00781-f021:**
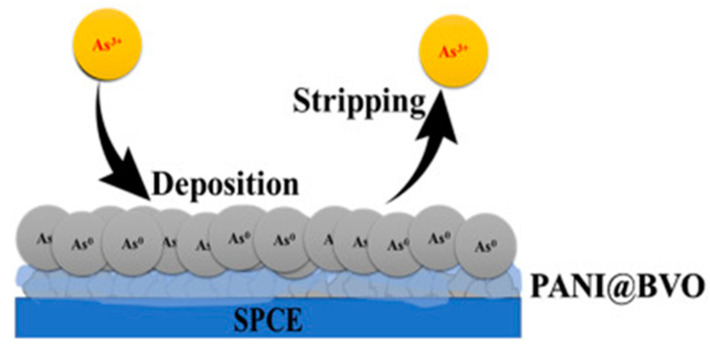
DPASV sensing mechanism of PANI@BiVO_4_ /SPCE for As^3+^ ions. (Reprinted with permission from [155], Copyright 2020, Elsevier B.V.)

**Figure 22 nanomaterials-12-00781-f022:**
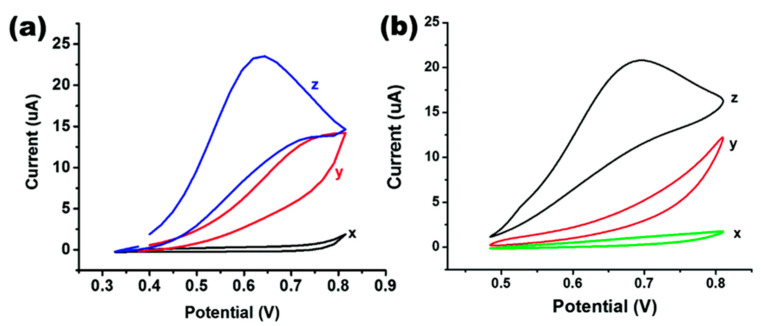
(**a**) CV curves of 50 ppm arsenic(III) ion standard under varying conditions for zirconia nanocubes, x: blank (black line), y: bare gold electrode (red line), and z: after fabrication (blue line); (**b**) CV curves of 50 ppm arsenic(III) ion standard under varying conditions for zirconia nanoparticles: x: blank (green line), y: bare gold electrode (red line), and z: after fabrication (black line). (Reprinted with permission from [156], Copyright 2016, Royal Society of Chemistry.)

**Figure 23 nanomaterials-12-00781-f023:**
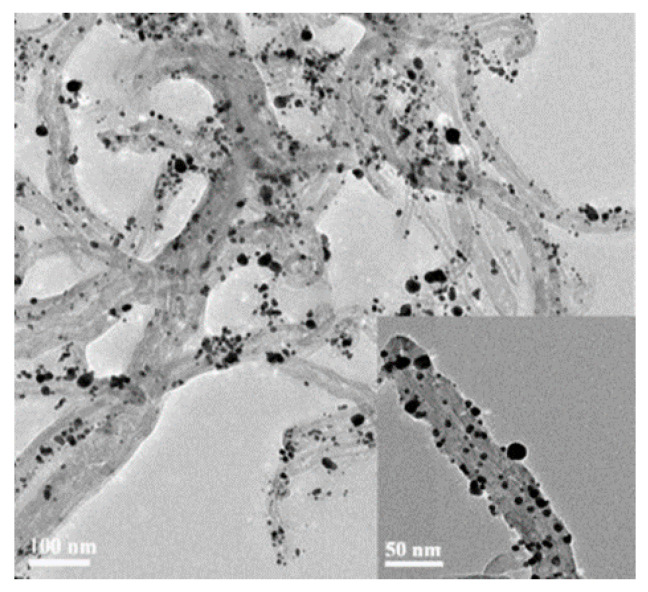
TEM photograph of Pt nano/CNTs synthesized by microwave irradiation; the inset is a higher magnification image of Pt/CNTs nanocomposite. (Reprinted with permission from [104], Copyright 2008, Elsevier B.V.)

**Figure 24 nanomaterials-12-00781-f024:**
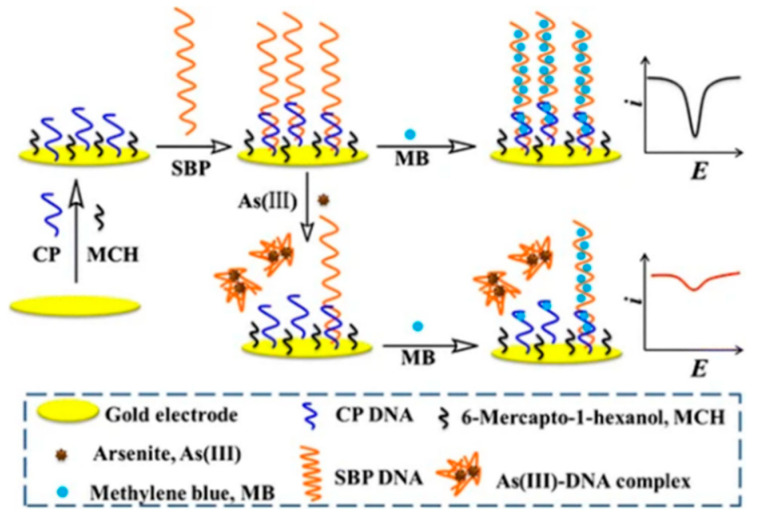
Schematic diagram of the fabrication of an electrochemical method for As(III) detection based on As(III)-induced DNA conformational changes and the electrochemical indicator MB. (Reprinted with permission from [200], Copyright 2017, Springer Nature Switzerland AG.)

**Figure 25 nanomaterials-12-00781-f025:**
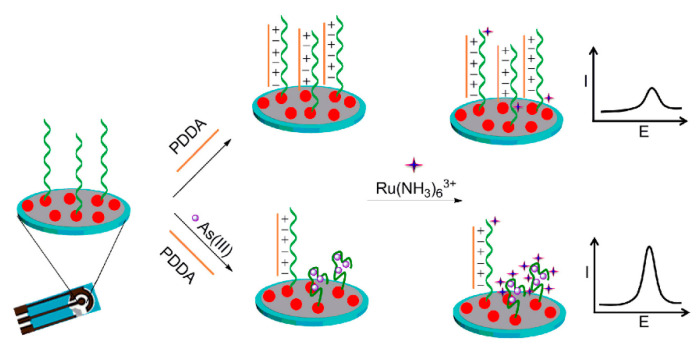
Schematic diagram of AuNPs/SPCE-based detection with Ars-3 aptamer. (Reprinted with permission from [203], Copyright 2016, Elsevier B.V.)

**Figure 26 nanomaterials-12-00781-f026:**
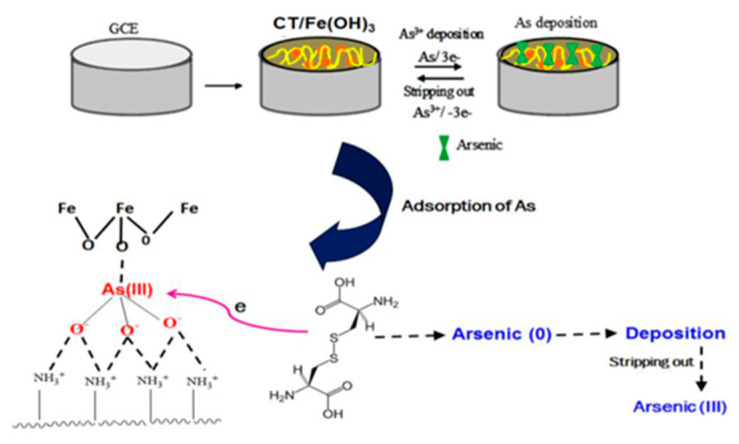
Schematic diagram of the scheme for the detection of arsenic ions by Fe–chitosan composites. (Reprinted with permission from [158], Copyright 2016, Elsevier B.V.)

**Figure 27 nanomaterials-12-00781-f027:**
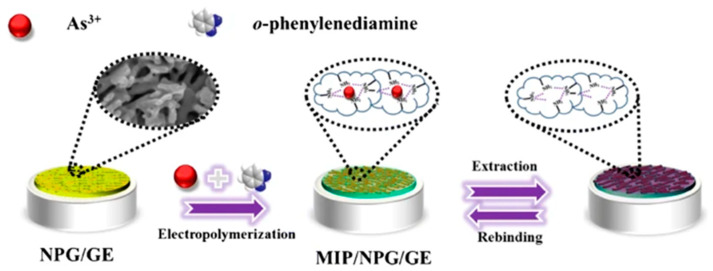
Schematic diagram of IIPs/NPG/GE formation process. (Reprinted with permission from [221], Copyright 2020, Springer Nature Switzerland AG.)

**Table 1 nanomaterials-12-00781-t001:** Comparison of gold electrodes.

Electrode	Method	Sensitivity (μA/ppb)	Linear Range(ppb)	LOD (ppb)	Reference
composite gold electrode	DPASV			0.32	[26]
gold side disk rotating electrode	DPASV		5–80		[27]
gold screen-printed electrode	SWASV	0.03	0–200	2.5	[24]
gold film	SCP ^1^			0.022	[28]
gold nanoelectrode ensembles	SWASV	3.14		0.02	[29]
gold nanofilm	LSV ^2^		0.2–375	0.04	[17]
Au(111)-like polycrystalline gold electrode	SWASV	0.097 μA ppb^−1^ cm^−2^	0–1123.8	0.28	[16]
vibrating gold microwire electrode	DPASV	0.014 μA ppb^−1^ mm^−1^ V^−1^	0.07–3.0	0.07	[21]
lateral gold electrode	ASV	9.15	0.1–15	0.06	[15]
MEA-modified Au electrode	DPASV	0.0366	0.2–300	0.02	[30]
Au-MEE	LSASV		1–10	0.09	[31]
Au disc electrode	ASV	18.69	0.75–299.68	0.075	[22]
nanoporous gold microelectrode	LSV	1.74 × 10^−4^	1.50–14,984	1.50	[32]
two gold electrodes	ASV		0.0374–0.7492	0.0097	[33]
Au-wire electrode	ASCP ^3^			0.42	[34]
porous gold electrode	SWV		0.1–14	0.1	[35]
gold electrode	ASV		0−5000	850	[36]
nanoporous gold microelectrode	SWASV	29.75 μA ppb^−1^ cm^−2^	2–30 and 10–200	0.62	[25]
gold wire microelectrode	ASV	6.8 μA ppb^−1^ cm^−2^		2.6	[23]
arsenite-selective ionophore film-Au	ASV		10–100	1.10	[37]
gold nanotextured electrode	ASV	39.54 μA ppb^−1^ cm^−2^	0.1–9	0.1	[38]
gold nanostar	SWSV ^4^		2.5–764.2	0.8	[39]

^1^ stripping chronopotentiometry; ^2^ linear sweep voltammetry; ^3^ anodic stripping chronopotentiometry; ^4^ square wave stripping voltammetry.

**Table 2 nanomaterials-12-00781-t002:** Gold-trimmed electrodes.

Electrode	Method	Sensitivity (μA/ppb)	Linear Range (ppb)	LOD (ppb)	Reference
Au-ITO	LSV			5	[55]
AuNPs	ASV	0.400 μA·V ppb^−1^	0.5–15	0.25	[40]
AuNPs-PANI	SWV			0.4	[56]
Au-coated boron-doped diamond thin-film	DPASV		0.01–40	0.005	[57]
AuCNT	ASV	26.49		0.1	[58]
3DAu nanodendrite network porous structure	DPSV		0.1–70	0.1	[47]
AuNPs/Fe_3_O_4_	SWASV	13.55	0.01–1	0.00097	[59]
Nafion-Ibu-AuNSs	CV		0.1–1800	0.018	[46]
citrate stabilized AuNPs	SWV		0.05–1 and 1–15	0.025	[60]
PDDA-AuNPs	DPV	0.017	0–7492	4.36	[61]
NF (Au nano)	SWV	0.32	0.1–12.0	0.047	[62]
ERGO-AuNPs	ASV	0.16	0.75–374.6	0.20	[63]
AuNP-SPE	LSAdSV ^1^	0.014	0.75–749.2		[64]
Au/Te	SWASV	6.35		0.0026	[65]
3D porous Au /TiO_2_	SWASV	0.064	7.49–599.36	3.00	[66]
CB-AuNPs/SPE	ASV	0.63	2–30	0.4	[67]
MnO_x_-AuNPs	LS-ASV	2.73 µA ppb^−1^ cm^−2^	0.5–80	0.057	[68]
AuNP	DPASV		4–1498	0.9	[42]
TTCN-AuNPs	CV		0.0019–2.55	0.0006	[69]
gold film	FI-DPASV		1.0–30	0.81	[45]
AuNPs	EME ^2^-ASV		0.5–10 and 10–600	0.18	[52]
AuNPs-C films	ASV	0.026	1–100	0.55	[70]
EG-AuNPs	SWASV			0.58	[71]
AuNPs-bt	CV		74.92–127,364	7.49	[53]
Au@Fe_3_O_4_-RTIL	SWASV	458.66 μA ppb^−1^ cm^−2^ /86.89 μA ppb^−1^ cm^−2^	0.1–1/1–10	0.0022	[72]
np-Au	SWASV	0.60	0.5–15	0.0315	[73]
MnFe_2_O_4_/Au hybrid nanospheres	SWASV	0.315		3.37	[74]
AuNpCµF	DPV			0.9	[75]
GCE-AuNPs	SWASV		10–12,000	0.15	[76]
AuNPs-PCWEs	AS-chronoamperometry	0.083		2.2	[77]
AuNPs-RGO	ASV	0.092	1–20	0.13	[78]
AuNPs-PpyNW	I-V	0.0029 and 0.0585	7.49–599.36 and 749.21–5244.51	23.97	[79]
Au/SiO_2_	SWV		0.1–40	0.07	[80]
ZrF-8CAu	CV		5–700	1	[81]
3D-rGO/AuNPs	EIS		3.8 × 10^−7^–3.0 × 10^−4^	1.4 × 10^−7^	[82]
SPE/CNF-CHIT@Au nano	FIA-ECD	0.2181	100–100,000	11.4	[83]
AuNP/BDD	SWASV		100–1500	20	[84]
AuNPs/CeO_2_-ZrO_2_	SWASV	0.976	0.5–15	0.137	[85]
AuNPs-SPCE	ASV	0.11	30–150	8.9	[41]
Buckypaper modified by GNP	LSASV		0.75–750	0.75	[86]
AuNP-Au film	ASLSV	0.027 μA ppb^−1^ cm^−2^	1–150	0.42	[87]
eAuNP-SPE	DPASV		0.5–20	0.22	[88]
gold film–plastic	DPSV		10–500	5	[89]
rGO-Au nano	SWASV		1.0–50.0	0.08	[90]
AuNPs/gC_3_N_4_	LSASV	3.07	0.375–74.921	0.22	[91]
3D NPG-ITO	DPASV	9.837	0.1–50	0.054	[92]
SiNPs/AuNPs	LSASV		10–100	5.6	[93]
MCPTH-AuNPs	CV			1	[94]
AuNS/SPCE	SWASV	0.2213	0–100	0.8	[49]
gold nanoparticles and crystal violet	DPV	0.075 μA ppb^−1^ cm^−2^	149.84–1648.24	59.94	[50]
gold nanostar	SWV	0.101	0–100	2.9	[95]
Au-PANI-Fe-CNF	SWASV	0.04	5–400	0.5	[54]
quasi-hexagonal gold nano	DPASV		0.075–30	0.11	[96]
GNR	DPASV		0.90–38.99	0.72	[48]
AuNP-rLA-Lcyst	SWV-ASV	0.1	3–25	3	[97]
AuNPs-Co_3_O_4_	SWASV	12.1/3.7	0.1–1/1–20	0.09/0.79	[98]
GO/Fe_3_O_4_@ PMDA/AuNPs	SWASV		5–500	0.15	[99]
AuNPs	SWSV	0.1007		16.73	[51]
Au nano/Fe-MOF	SWASV	4.708	2–30	0.0085	[100]
GC-AuNP-ArOx	CV	46.05	0.75–749.21	0.37	[101]

^1^ Linear sweep adsorptive stripping voltammetry; ^2^ electromembrane extraction.

**Table 3 nanomaterials-12-00781-t003:** Other precious-metal-modified electrodes.

Electrode	Method	Sensitivity (μA/ppb)	Linear Range (ppb)	LOD (ppb)	Reference
Pt nano	ASV	2.94 × 10^−3^	74.92–3746.08	2.1	[121]
Pt nano	SWV		0–100	0.5	[107]
PtNTAEs	LSV	0.011	749.21–14,984.32	0.1	[103]
Pt nano/CNTs	LSV	9.34 × 10^−3^	374.61–74,921.6	0.12	[104]
Pt nano	CV			5.68	[122]
Gr-nPt	SWASV		0.75–7.49	0.082	[106]
Pt nano (2.3 nm)	SWASV	0.356	0–1000		[110]
Nafion/Pt	ASV	0.036	0–40	<10	[102]
Pt nano	ASV		0–100	16.50	[105]
Pt_1_ /MoS_2_	SWASV	3.31	0.5-8	0.05	[109]
Pt nano	ASV	6.3 × 10^−7^ C μM^−1^	3.75–74.92	4	[123]
SWE	LSASV			0.09	[111]
AgNPs/CT	DPASV		10–100	1.20	[113]
AgNPs-GO	ASV	2.41	1–28.11	0.018	[114]
Ag-GCE	SWASV	0.98	10–60	4.2	[124]
Ag-SPCE	SWASV	0.6	10–80	8.4	[124]
AgNPs	DPASV		3.75–14.98	1.03	[112]
GSH/DTT/Asn-Ag NPs	DPV		0.01–40	5.2 × 10^−3^	[115]
IrOx-BDD	chronoamperometry	0.056	1.50–3746.08	0.15	[125]
Ir-BDD	CV	1.24 × 10^−3^ μA ppb^−1^ cm^−2^		1.5	[126]
IrO_2_ nanotubes	DPV		0–5993.73	7.49	[117]
Au-IrM	SWASV	3.19 × 10^−4^/2.64 × 10^−3^	0.75–3.75/0.07–0.75	0.037	[127]
Ir-BDD	CV	7.47 × 10^−4^ μA ppb^−1^ cm^−2^	74.92–7492.16	347.63	[116]
Ru NPs	DPV	2.38 × 10^−3^		0.1	[118]
[Ru(bpy)_3_]^2+^-GO	DPV	0.32	7.49–14.98	0.015	[128]
Ru(II)-tris(bipy)-GO	CV	1.42 μA ppb^−1^ cm^−2^	3.75–59.93	2.25	[129]
Pd-PEDOT	DPASV	19.78 μA ppb^−1^ cm^−2^	0–749.21	0.52	[119]
Pt–Pd_sds_	SWV		74.92–16,857.36	0.2	[120]

**Table 4 nanomaterials-12-00781-t004:** Bimetallic modified electrodes.

Electrode	Method	Sensitivity (μA/ppb)	Linear Range (ppb)	LOD (ppb)	Reference
Au-Pd NPs	SWV		1–25	0.25	[131]
Au-Pd NPs	SWASV	3.9		0.024	[132]
Au/Te crystals	SWASV	6.35	0.1–10	0.0026	[65]
Au-Pt NPs	LSASV		0.37–224.76	0.28	[133]
Au-Cu	SWASV	1.63 μA ppb^−1^ cm^−2^		2.09	[134]
Au-PtNPs/PANI	SWASV	0.23	2.47–14.98	1.48	[135]
C-AuNPs	SWV		0.5–100	0.092	[136]
Ag-Au	CV/DPV		0.01–10	0.003 × 10^−3^	[137]
Fe_3_O_4_-Au	SWASV	122	1–100	0.22	[138]
FePt	SWV	0.42	1–5	0.8	[130]
Pt-Fe	ASV	0.064		0.75	[139]

**Table 5 nanomaterials-12-00781-t005:** Other metal- and compound-modified electrodes.

Electrode	Method	Sensitivity (μA/ppb)	Linear Range (ppb)	LOD (ppb)	Reference
Au NPs/Fe_3_O_4_	SWV	13.55	0.01–1	0.00097	[59]
Fe-CB/PE	ASV		0.4–20	0.16	[157]
Fe_3_O_4_-RTIL	SWASV	4.91	1–10	0.0008	[144]
Chitosan-Fe(OH)_3_	ASV	8.39	2–100	0.072	[158]
Au @Fe_3_O_4_-RTIL	SWASV	58.66 μA ppb^−1^ cm^−2^	0.1–1	0.0022	[72]
rGO/Fe_3_O_4_		1.922	2–20	0.3	[159]
rGO-Fe_3_O_4_	DPASV		2–300	0.10	[143]
rGO/Fe_3_O_4_	SWASV	0.281		0.12	[141]
Fe-MOF @mFe_3_O_4_@mC	EIS		7.49 × 10^−4^–0.75	5.04 × 10^−4^	[160]
Fe Pc/Si-NP	DPASV	0.20		3.66	[161]
CoPc/Si-NP	DPASV	0.18		4.39	[161]
Au/Fe_3_O_4_	SWASV	9.43	0.1–10	0.0215	[162]
α-FeOOH	CV		0.75–1498.43	0.37	[163]
Fe_2_V_4_O_13_–polypyrrole	DPASV		0–500	0.3	[164]
Fe_3_O_4_-Ag/Au HNSs-rGONs	CV	52	0.1–20	0.01	[165]
CN-wrapped IL-modified ZF-Ms (CN@ZF-Ms-IL)	SWASV	41.08	1–60	0.0006	[166]
Fe_3_O_4_–rGO	SWV	2.15	1–20	1.19	[142]
GO/Fe_3_O_4_@PMDA/AuNPs	SWSV		0.5–750	0.15	[99]
Fe_3_O_4_-Au-IL	SWASV	122	1–100	0.22	[138]
Fe_3_O_4_/Co_3_S_4_	SWASV	4.359	1.0–10.0	0.691	[145]
MnO_x_/AuNPs	LS-ASV	2.73 µA ppb^−1^ cm^−2^	0.5–80	0.057	[68]
MnFe_2_O_4_ NCs	SWASV	0.295		1.95	[167]
rGO/MnO_2_	SWASV	0.175	0.1–50	0.05	[168]
AuNPs/α-MnO_2_	SWASV	0.828	1–10	0.019	[169]
MnFe_2_O_4_/Au hybrid nanospheres	SWASV	0.315		3.37	[74]
Mn_2_O_3_/CeO_2_	SWASV	0.0414		3.35	[146]
Nafion/α-MnO@PDA	SWASV	0.13	10–150	3.2	[170]
MnO_2_ /POT/rGO	DPV	0.00163	0.01–0.9	0.042	[147]
Bi-NPs	SWV			5	[153]
EG-Bi	SWASV			0.014	[154]
PANI@BiVO_4_	DPASV	6.06 μA ppb^−1^ cm^−2^	0.01–300	0.0072	[155]
F-doped CdO thin films	CV	5.747 × 10^−3^	4.55–41	0.00455	[171]
CoOx	CV	1.49 × 10^−3^		0.82	[148]
Co-rGO	ASV			0.31	[172]
AuNPs-Co_3_O_4_	SWASV	12.1/3.7	0.1–1/1–20	0.09/0.79	[98]
Fe_3_O_4_ /Co_3_S_4_	SWASV	4.359	1.0–10.0	0.691	[145]
In_0.38_Ga_0.62_N/Si(111)	SWV		10–50	9.27	[173]
Pt_1_ /MoS _2_	SWASV	3.31	0.5-8	0.05	[109]
PbO_2_/rGO				0.75	[174]
Au-ITO	ASV			5	[175]
SnO_2_ nanosheets	SWASV	0.058	5–300	4.6	[150]
Nafion/SnO_2_ nanoneedles	CV	28.13 μA ppb^−1^ cm^−2^	50–500	10	[151]
3D NPG-ITO	DPASV	9.837	0.1–50	0.054	[92]
Gemini-ITO	SWV		1–100	0.88	[176]
SrTiO_3_ /β-CD	Amperometry	0.0053 μA μM cm^−2^	749.21–10,489.02	1.50	[152]
CP-ThO_2_ NP	DPASV	0.54	3–180	0.1	[177]
3D porous Au/TiO_2_	SWASV	0.064	7.49–599.36	3.00	[66]
TiO_2_-GSE	LSV	1.10	10–80	10	[178]
ZrO_2_-nanocubes	CV		5–60	5	[156]
ZrF-8CAu	CV		5–700	1	[81]
Zr-G-PGE (As(V))	DPV	1.36	0.10–40.0	0.12	[179]
AuNPs/CeO_2_-ZrO_2_	SWASV	0.976	0.5–15	0.137	[85]

**Table 6 nanomaterials-12-00781-t006:** Carbon material-modified electrode.

Electrode	Method	Sensitivity (μA/ppb)	Linear Range (ppb)	LOD (ppb)	Reference
ERGO-AuNPs	ASV	0.16	0.75–374.61	0.20	[63]
Gr-nPt	SWASV		0.75–7.5	0.082	[106]
Au/GO/Leucine/Nafion	CV	0.03 μA ppb^−1^ cm^−2^		500	[181]
NH_2_-GO	SWASV	130.631 μA ppb^−1^ cm^−2^		0.162	[182]
3D-rGO/AuNPs	EIS		3.8 × 10^−7^–3.0 × 10^−4^	1.4 × 10^–7^	[82]
[Ru(bpy)_3_]^2+^-GO	DPV		6.00–1123.83	1.57	[183]
Ru(II)-tris(bipy)-GO	CV	1.42 μA ppb^−1^ cm^−2^	3.75–59.94	2.25	[129]
TTCA/rGO	SWASV		0–10	0.054	[184]
Gr/MOF	DPASV		0.2–25	0.06	[185]
RM-rGO	SWASV	2.49		0.07	[186]
SPGE	DPAV		0.0–5.0	0.28	[187]
Au-PANI-Fe-CNF	SWASV	0.04	5–400	0.5	[54]
SH-SWCNTs	LSV	1.33		0.008	[188]
DNA–SWCNT	LSV	0.17	0–33.6	0.05	[189]
Pt nano/CNTs	LSV	9.34 × 10^−3^	374.61–74,921.6	0.12	[104]
CNTs/Leucine/Nafion	CV	0.27	0.37–149.84	0.12	[190]
CNTs/Nafion/Glutamine		1.33	0.075–37.835	2.72	[191]
CNTs-GNPs	LSASV	135		0.5	[192]
ssDNA/SWCNTs	DPV		0.5–10	0.5	[193]
Eu-MGO/Au@MWCNT	SWSV		0.99–100.0	0.27	[194]
Buckypaper modified by GNP	LSASV		0.75–750	0.75	[86]
CQDs/f-MWCNTs/GO	DPV		7.49 × 10^−3^–0.82	0.037	[195]
MMWCNTs-D-NH_2_	SWASV	0.5613	1.0–50.0	0.46	[196]

**Table 7 nanomaterials-12-00781-t007:** Biomolecule-modified electrodes.

Electrode	Method	Sensitivity (μA/ppb)	Linear Range (ppb)	LOD (ppb)	Reference
SBP DNA	DPV		0.1–200	0.075	[200]
(GT)_21_-ssDNA and PB@GO	DPV		0.2–500	0.058	[206]
GH-APTES-Fe_3_O_4_ NP	SWV	1.92/0.12	13.3–65.8/117–241	1.6	[207]
SAMs	ASV		2–40	0.5	[208]
L-tryptophan	SWASV		7.49 × 10^−3^–7.49	0.90 × 10^−3^	[209]
AuNP-rLA-Lcyst	SWV-ASV	0.1	3–25	3	[97]
*P. cruentum*	DPASV		2.5–20	1.08	[210]
MTs	ASV		5–1000	13	[211]
AgNPs/CT	DPASV		10–100	1.20	[113]
Chitosan-Fe(OH)_3_	ASV	8.39	2–100	0.072	[158]
SPE/CNF-CHIT@Au nano	FIA-ECD	0.2181	100–100,000	11.4	[83]
ACh-SPC	chronoamperometry			2689.63	[212]
HCR and RecJf exonuclease	EIS		0.1–500	0.02	[213]
AF /AuNPs-SPCE	CV		0.500–1999.61	137.85	[214]
GC-AuNP-ArOx	CV	46.05	0.75–749.21	0.37	[101]
*Escherichia coli*	CV		0.94–3.75/3.75–30	0.8	[215]
Whole-Cell Biosensors	CV		0–100		[216]
*E. coli*	CV	0.122		1.5	[217]
Ars-3	DPV		0.015–7.49	0.011	[203]
3D-rGO/Au NPs/ssDNA	EIS		3.8 × 10^−7^–3.0 × 10^−4^	1.4 × 10^−7^	[82]
ArsSApt	EIS		50–10,000	59.94	[218]
C-AuNPs	SWV		0.5–100	0.092	[136]
Ars-3/AuNPs-GO-MB	DPV		0.4–10,000	0.2	[197]

## Data Availability

Not applicable.

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
