# Peer review of "Advances in Electrochemical Detection Electrodes for As(III)"

_nanomaterials, 2022, doi:10.3390/nano12050781_

Round 1
Reviewer 1 Report
The work is well organized, well written. The references are appropriate,
relevant and rich. The work can be useful to those who work in the field.
I recommend its publication in present form.
The theme of the paper is important and topical for human health. the authors made a very detailed and thorough review of the state of the art of electrochemical sensing methods for arsenic ions based on nanomaterial-modified electrodes. In my opinion, a review work must present what is found in the literature, clarify the strengths or disadvantages of the different techniques, in order to be a starting point and a reference for those who want to continue their studies in the sector. I think the authors have achieved their goal. I especially appreciate the tables in which the salient data of the different solutions in literature are reported, because they allow the reader to make a comparison among different solutions quickly and clearly.
Reviewer 2 Report
The reviewer understand the authors huge efforts. Before its publication, minor revisions are expected to be effective.
1) Abstract had an introduction but had no contents. By shortening the background, the brief summary should be written.
2) The detection limit was given by various units. Various values should be transformed to those in one common unit, and should be added.
3) How was the influence of the sample preparation, such as solvent, anion and interfering materials?
4) Based on many methods and data which were collected and arranged, their advantages and disadvantages should be discussed by comparing with each other. In addition to a brief comment of advantages given in Summary, their detail should be discussed at the last part of the manuscript. Could the authors prepare table showing the evaluation of operation, sensitivity, selectivity, rapid detection?
Reviewer 3 Report
This paper is a review of electrochemical sensing methods for arsenic ions based on nanomaterial-modified electrodes. The manuscript is very well written but need a minor corrections. There are mainly stylistic remarks.
- The abstract needs to be improved, most of it is about Arsenic, while the whole content of the paper is described with only one last sentence.
- The text in Figure 1 is too small.
3. In lines 141-142 commas are used incorrectly, there is lack of space before references. - In table 1, the abbreviations: GNEE, (PAu) electrode and AuNS-CPSPE should be explained.
Moreover, the units (5×10-10-1×10-8M 1.3×10-10M) should be changed to nano (nM).
- In line 216 there is missing a space (0.939[51] .)
In line 224 there is missing a space (has been reported[53],).
In line 290 there is missing a space (8M).
In line 340 there is missing a space (conditions[119]).
In line 459 there is missing a space (fabricated[146]).
In line 563 there is missing a space (performance[85]).
In lines 628-629 there is missing a space between citations (metals[185],multi- 628walled carbon nanotubes[195],precious metals[114] and biomolecules[197] among others.).
In line 767 there is missing a space, and one too much after bracket (curve[221] .).
In line 781 the word "Second" should be changed to "second".
- The units (1.38×10-8M) should be changed to nano (nM) in table 3.
- In line 439 there is problem in subscript (Fe3O4-RTIL).
- In line503 there is problem with dot.
- In table 5 the value: 4.55 × 10-3 should be changed to 0.00455.
